# Dynamic Conditional Optimal Transport through Simulation-Free Flows

**Gavin Kerrigan**
Department of Computer Science
University of California, Irvine
gavin.k@uci.edu

**Giosue Migliorini**
Department of Statistics
University of California, Irvine
gmiglior@uci.edu

**Padhraic Smyth**
Department of Computer Science
University of California, Irvine
smyth@ics.uci.edu

## Abstract

We study the geometry of conditional optimal transport (COT) and prove a dynamic formulation which generalizes the Benamou-Brenier Theorem. Equipped with these tools, we propose a simulation-free flow-based method for conditional generative modeling. Our method couples an arbitrary source distribution to a specified target distribution through a triangular COT plan, and a conditional generative model is obtained by approximating the geodesic path of measures induced by this COT plan. Our theory and methods are applicable in infinite-dimensional settings, making them well suited for a wide class of Bayesian inverse problems. Empirically, we demonstrate that our method is competitive on several challenging conditional generation tasks, including an infinite-dimensional inverse problem.

## 1 Introduction

Many fundamental tasks in machine learning and statistics may be posed as modeling a conditional distribution $\nu(u \mid y)$ where obtaining an analytical representation of $\nu(u \mid y)$ is impractical. While sampling-based approaches such as Markov Chain Monte Carlo (MCMC) methods are useful, they suffer from several limitations. First, MCMC requires numerous likelihood evaluations, rendering it prohibitively expensive in scientific and engineering applications where the likelihood is determined by an expensive numerical simulator. Second, MCMC must be run anew for every observation $y$, which is impractical in applications such as Bayesian inverse problems [Dashti and Stuart, 2013] and generative modeling [Mirza and Osindero, 2014]. These limitations motivate the need for a *likelihood-free* [Cranmer et al., 2020] and *amortized* [Amos et al., 2023] approach. While methods like ABC [Beaumont, 2010] and variational inference [Blei et al., 2017] partially address these challenges, they are difficult to scale to high dimensions or have limited flexibility.

Recently, generative models such as normalizing flows [Papamakarios et al., 2019, 2021], GANs [Ramesh et al., 2022], and diffusion models [Sharrock et al., 2022] have shown promise in amortized and likelihood-free inference. These models may be viewed in the framework of *measure transport* [Baptista et al., 2020], where samples $u \sim \eta(u)$ from a tractable source distribution are transformed by a mapping $T(y, u)$ such that the transformed samples are approximately distributed as $\nu(u \mid y)$. One way to achieve this is through *triangular* mappings [Baptista et al., 2020, Spantini et al., 2022], where a joint source distribution $\eta(y, u)$ is transformed by a mapping of the form $T : (y, u) \mapsto (T_Y(y), T_U(y, u))$. Under suitable assumptions, if $T$ transforms the source $\eta(y, u)$ into the target $\nu(y, u)$, then $T_U(y, -)$ couples the conditionals $\eta(u \mid y)$ and $\nu(u \mid y)$.

38th Conference on Neural Information Processing Systems (NeurIPS 2024).

Typically, such a map $T$ is not unique [Wang et al., 2023], and a natural idea is thus to regularize the transport and search for an admissible mapping that is in some sense optimal. In other words, learning a conditional sampler may be phrased as finding a conditional optimal transport (COT) map. While there exists some work on learning COT maps, these approaches often rely on a difficult adversarial optimization problem [Baptista et al., 2020, Hosseini et al., 2023, Bunne et al., 2022, Ray et al., 2022] or simulating from the model during training [Baptista et al., 2023, Wang et al., 2023]. In this work, we propose a conditional generative model for likelihood-free inference based on a dynamic formulation of conditional optimal transport. Specifically, our contributions are as follows:

1. We develop a general theoretical framework for dynamic conditional optimal transport in separable Hilbert spaces. Our framework is applicable in infinite-dimensional spaces, enabling applications in function space Bayesian inference. In Section 4, we study the *conditional Wasserstein space* $\mathbb{P}_p^\mu(Y \times U)$ and show that this space admits constant speed geodesics between any two measures. In Section 5, we characterize the absolutely continuous curves of measures in $\mathbb{P}_p^\mu(Y \times U)$ via the continuity equation and *triangular* vector fields. As a consequence, we obtain conditional generalizations of the McCann interpolants [McCann, 1997] and the Benamou-Brenier Theorem [Benamou and Brenier, 2000].

2. In Section 6, we propose COT flow matching (COT-FM), a simulation-free flow-based model for conditional generation. This model directly leverages our theoretical framework, where we learn to model a path of measures interpolating between an arbitrary source and target distribution via a geodesic in the conditional Wasserstein space.

3. In Section 7, we demonstrate our method on several challenging conditional generation tasks. We apply our method to two Bayesian inverse problems – one arising from the Lotka-Volterra dynamical system, and an infinite-dimensional problem arising from the Darcy Flow PDE. Our method shows competitive performance against recent COT methods.

## 2   Related Work

**Conditional Optimal Transport.**   Conditional Optimal Transport (COT) remains relatively under-explored in both machine learning and related fields. Recent approaches learn static COT maps via input convex networks [Bunne et al., 2022, Wang et al., 2023] or normalizing flows [Wang et al., 2023]. In addition, there have been a number of heuristic approaches to conditional simulation through W-GANs [Sajjadi et al., 2017, Adler and Öktem, 2018, Kim et al., 2022, 2023], for which Chemseddine et al. [2023] provide a rigorous basis. Closely related to our work are those which employ triangular plans [Carlier et al., 2016, Trigila and Tabak, 2016], which have been modeled through GANs in Euclidean spaces [Baptista et al., 2020] and function spaces [Hosseini et al., 2023]. In contrast, we use a novel *dynamic* formulation of COT, which we model through a generalization of flow matching [Lipman et al., 2022, Albergo et al., 2023b, Liu et al., 2022]. This allows us to use flexible architectures while avoiding the difficulties of training GANs [Arora et al., 2018].

**Simulation-Free Continuous Normalizing Flows.**   Flow matching [Lipman et al., 2022] (and the closely related stochastic interpolants [Albergo et al., 2023a] and rectified flows [Liu et al., 2022]) are a class of methods for building continuous-time normalizing flows in a simulation-free manner. Notably, these works do not approximate an optimal transport between the source and target measures. Pooladian et al. [2023] and Tong et al. [2023] propose instead to couple the source and target distributions via optimal transport, leading to marginally optimal paths. In this work, we study an extension of these techniques for conditional generation.

While some works [Davtyan et al., 2023, Gebhard et al., 2023, Isobe et al., 2024, Wildberger et al., 2024] have applied flow matching for conditional generation, these approaches do not employ COT. Notably, these approaches are limited to the finite-dimensional setting, whereas our method adds to the growing literature on function-space generative models [Hosseini et al., 2023, Kerrigan et al., 2023, 2024, Lim et al., 2023, Franzese et al., 2024]. Concurrent work by Chemseddine et al. [2024] develops the foundation of dynamic COT with applications to flow matching, and concurrent work by Barboni et al. [2024] develop the theory of dynamic COT for the purposes of studying infinitely deep ResNets. However, these works focus on the Euclidean setting, whereas our methods are applicable in general separable Hilbert spaces.

# 3 Background and Notation

Let $X, X'$ represent arbitrary separable Hilbert spaces, equipped with the Borel $\sigma$-algebra. We use $\mathbb{P}(X)$ to represent the space of Borel probability measures on $X$, and $\mathbb{P}_p(X) \subseteq \mathbb{P}(X)$ to represent the subspace of measures having finite $p$th moment. If $\eta \in \mathbb{P}(X)$ is a probability measure on $X$ and $T : X \to X'$ is measurable, then the pushforward measure $T_\# \eta(-) = \eta(T^{-1}(-))$ is a probability measure on $X'$. Maps of the form e.g. $\pi^X : X \times X' \to X$ represent the canonical projection.

We assume that we have two separable Hilbert spaces of interest. The first, $Y$, is a space of observations, and the second, $U$, is a space of unknowns. These spaces may be of infinite dimensions, but a case of practical interest is when $Y$ and $U$ are finite dimensional Euclidean spaces. We will consider the product space $Y \times U$, equipped with the canonical inner product obtained via the sum of the inner products on $Y$ and $U$, under which the space $Y \times U$ is also a separable Hilbert space. Let $\eta \in \mathbb{P}(Y \times U)$ be a joint probability measure. The measures $\pi^Y_\# \eta \in \mathbb{P}(Y)$ and $\pi^U_\# \eta \in \mathbb{P}(U)$ obtained via projection are the *marginals* of $\eta$. We use $\eta^y \in \mathbb{P}(U)$ to represent the measure obtained by conditioning $\eta$ on the value $y \in Y$. By the disintegration theorem [Bogachev and Ruas, 2007, Chapter 10], such conditional measures exist and are essentially unique, in the sense that there exists a Borel set $E \subseteq Y$ with $\pi^Y_\# \eta(E) = 0$, and the $\eta^y$ are unique for $y \notin E$.

## 3.1 Static Conditional Optimal Transport

In conditional optimal transport, we are given a target measure $\nu \in \mathbb{P}(Y \times U)$ and some source measure $\eta \in \mathbb{P}(U)$. We seek a transport map $T : Y \times U \to U$ such that, for any given $y \in Y$, the mapping $T(y, -) : U \to U$ transforms the source distribution $\eta$ into the conditional distribution $\nu^y$, i.e., $T(y, -)_\# \eta = \nu^y$. In a sense, $T$ can be thought of as a collection of transport maps indexed by $y \in Y$. If such a $T$ were available, by drawing samples $u_0 \sim \eta$ and transforming them, one would obtain samples $T(y, u_0) \sim \nu^y$. Solving this transport problem for each fixed $y$ is expensive at best, or impossible when only has a single (or no) samples $(y, u) \sim \nu$ for any given $y$. Thus, one must leverage information across different observations $y$. To that end, recent work has focused on the notion of *triangular mappings* $T : Y \times U \to Y \times U$ [Hosseini et al., 2023, Baptista et al., 2020] of the form $T(y, u) = (T_Y(y), T_U(T_Y(y), u))$ for some $T_Y : Y \to Y$ and $T_U : Y \times U \to U$. Triangular mappings are of interest as they allow us to obtain conditional couplings from joint couplings.

**Proposition 1** (Theorem 2.4 [Baptista et al., 2020], Prop. 2.3 [Hosseini et al., 2023])
*Suppose $\eta, \nu \in \mathbb{P}(Y \times U)$ and $T : Y \times U \to Y \times U$ is triangular. If $T_\# \eta = \nu$, then $T_U(T_Y(y), -)_\# \eta^y = \nu^{T_Y(y)}$ for $\pi^Y_\# \eta$-almost every $y$.*

In many scenarios of practical interest, the source measure $\eta$ and the target measure $\nu$ have the same $Y$-marginals. We will henceforth make this assumption, and use $\mu = \pi^Y_\# \eta = \pi^Y_\# \nu$ to represent this marginal. In this case, we may take $T_Y$ to be the identity mapping, so that the conclusion of Proposition 1 simplifies to $T_U(y, -)_\# \eta^y = \nu^y$ for $\mu$-almost every $y$. We note that in situations where such an assumption does not hold, one may simply preprocess the source measure $\eta$ via an invertible mapping $T_Y$ satisfying $[T_Y]_\# [\pi^Y_\# \eta] = \pi^Y_\# \nu$ [Hosseini et al., 2023, Prop 3.2].

Given a source and target measures $\eta, \nu \in \mathbb{P}^\mu(Y \times U)$ and a cost function $c : (Y \times U)^2 \to \mathbb{R}$, the *conditional Monge problem* seeks to find a triangular mapping solving

$$\inf_T \left\{ \int_{Y \times U} c(y, u, T(y, u)) \, \mathrm{d}\eta(y, u) \mid T_\# \eta = \nu, T : (y, u) \mapsto (y, T_U(y, u)) \right\}. \tag{1}$$

The conditional Monge problem also admits a relaxation under which one only considers couplings whose $Y$-components are almost surely equal. To that end, for $\eta, \nu \in \mathbb{P}^\mu_p(Y \times U)$ we define the set of *triangular couplings* $\Pi_Y(\eta, \nu)$ to be the couplings of $\eta$ and $\nu$ that almost surely fix the $Y$-components,

$$\Pi_Y(\eta, \nu) = \left\{ \gamma \in \mathbb{P}\left( (Y \times U)^2 \right) \mid \pi^{1;2}_\# \gamma = \eta, \pi^{3,4}_\# \gamma = \nu, \pi^{1,3}_\# = (I, I)_\# \mu \right\}. \tag{2}$$

In other words, a triangular coupling $\gamma \in \Pi_Y(\eta, \nu)$ has samples $(y_0, u_0, y_1, u_1) \sim \gamma$ such that $y_0 = y_1$ almost surely. The *conditional Kantorovich problem* seeks a triangular coupling solving

$$\inf_\gamma \left\{ \int_{(Y \times U)^2} c(y_0, u_0, y_1, u_1) \, \mathrm{d}\gamma(y_0, u_0, y_1, u_1) \mid \gamma \in \Pi_Y(\eta, \nu) \right\}. \tag{3}$$

Hosseini et al. [2023] prove the existence of minimizers to the conditional Kantorovich and Monge problems under very general assumptions. Moreover, optimal couplings to the conditional Kantorovich problem induce optimal couplings for $\mu$-almost every conditional measure. We refer to Appendix B and Hosseini et al. [2023] for further details.

## 4  Conditional Wasserstein Space

Motivated by our discussion on triangular transport maps, we introduce the conditional Wasserstein spaces, consisting of joint measures with finite $p$th moments and having fixed $Y$-marginals $\mu$. Interestingly, Gigli [2008, Chapter 4] studies the same space for the purposes of constructing geometric tangent spaces in the usual Wasserstein space.

**Definition 1** (Conditional Wasserstein Space)
*Suppose $\mu \in \mathbb{P}(Y)$ is given and $1 \le p < \infty$. The conditional p-Wasserstein space is*

$$\mathbb{P}_p^\mu(Y \times U) = \left\{ \gamma \in \mathbb{P}_p(Y \times U) \mid \pi_\#^Y \gamma = \mu \right\}. \tag{4}$$

We now equip $\mathbb{P}_p^\mu(Y \times U)$ with a metric $W_p^\mu$, the conditional Wasserstein distance. Intuitively, the conditional Wasserstein distance measures the usual Wasserstein distance between all of the conditional distributions in expectation under the fixed $Y$-marginal $\mu$.

**Definition 2** (Conditional $p$-Wasserstein Distance)
*Suppose $\eta, \nu \in \mathbb{P}_p^\mu(Y \times U)$ and $1 \le p < \infty$. The function $W_p^\mu : \mathbb{P}_p^\mu(Y \times U) \times \mathbb{P}_p^\mu(Y \times U) \to \mathbb{R}$,*

$$W_p^\mu(\eta, \nu) = \left( \mathbb{E}_{y \sim \mu} \left[ W_p^p(\eta^y, \nu^y) \right] \right)^{1/p} = \left( \int_Y W_p^p(\eta^y, \nu^y) \, \mathrm{d}\mu(y) \right)^{1/p} \tag{5}$$

*is the conditional p-Wasserstein distance. $W_p$ is the usual Wasserstein distance for measures on $U$.*

By Jensen's inequality we have $W_p^\mu(\eta, \nu) \ge \mathbb{E}_{y \sim \mu} \left[ W_p(\eta^y, \nu^y) \right]$. In the following, we show that the conditional Wasserstein distance is a well-defined metric as well as a few other metric properties.

**Proposition 2** (Some Properties of $W_p^\mu$)
*Let $1 \le p < \infty$.*

  (a) *$W_p^\mu$ is well-defined, finite, and equals the minimal conditional Kantorovich cost.*

  (b) *$W_p^\mu$ is a metric on the space $\mathbb{P}_p^\mu(Y \times U)$.*

  (c) *There does not exist $C > 0$ such that $W_p^\mu(\eta, \nu) \le C W_p(\eta, \nu)$ for all $\eta, \nu \in \mathbb{P}_p^\mu(Y \times U)$.*

  (d) *For all $\eta, \nu \in \mathbb{P}_p^\mu(Y \times U)$, $W_p\left( \pi_\#^U \eta, \pi_\#^U \nu \right) \le W_p^\mu(\eta, \nu)$ and $W_p(\eta, \nu) \le W_p^\mu(\eta, \nu)$.*

Proposition 2(c, d) together shows that one should expect the topology generated by $W_p^\mu$ to be stronger than the unconditional distance $W_p$. Here, we note that Gigli [2008] and Chemseddine et al. [2023] previously showed that $W_p^\mu$ is a metric through an equivalence with restricted couplings. Our approach builds on the results of Hosseini et al. [2023] and is somewhat more direct, and hence our proofs may be of independent interest.

For the sake of concreteness, we include an example where the conditional 2-Wasserstein distance may be explicitly computed. Here, the necessary calculations follows from the fact that the conditional distributions of a multivariate are again Gaussian, and Gaussian distributions admit a closed-form expression for the usual unconditional 2-Wasserstein distance.

**Example: Gaussian Measures.**    Suppose $Y = U = \mathbb{R}$ and that $\eta, \nu \in \mathbb{P}_p^\mu(Y \times U)$ are Gaussians

$$\eta = \mathcal{N}(0, I) \qquad \nu = \mathcal{N}\left( 0, \begin{bmatrix} 1 & \rho \\ \rho & 1 \end{bmatrix} \right) \qquad |\rho| < 1. \tag{6}$$

It follows that $\mu = \pi_\#^Y \eta = \pi_\#^Y \nu = \mathcal{N}(0, 1)$. As $\eta^y, \nu^y$ are Gaussians, their $W_2$ distance admits a closed form and we can directly compute the expectation in Equation (5) to obtain $W_2^{\mu,2}(\eta, \nu) = 2(1 - \sqrt{1 - \rho^2})$. This is zero if and only if $\rho = 0$, i.e. $\eta = \nu$. However, $\pi_\#^U \eta = \pi_\#^U \nu = \mathcal{N}(0, 1)$ and $W_2(\pi_\#^U \eta, \pi_\#^U \nu) = 0$ regardless of $\rho$. Moreover, the unconditional distance is $W_2^2(\eta, \nu) = 2\left( 2 - \sqrt{1 - \rho} - \sqrt{1 + \rho} \right)$, from which it is easy to verify that $W_2(\eta, \nu) \le W_2^\mu(\eta, \nu)$. See Appendix C for a similar derivation which applies to arbitrary Gaussians.

**Conditional Wasserstein Space as a Geodesic Space.** We now turn our attention to the geodesics in $\mathbb{P}_p^\mu(Y \times U)$. In particular, we show that there exists a constant speed geodesic between any two measures in $\mathbb{P}_p^\mu(Y \times U)$, generalizing a similar result in the unconditional setting [Santambrogio, 2015, Theorem 5.27]. Moreover, we show that under suitable regularity assumptions, solutions to the conditional Monge problem (1) induce constant speed geodesics. Our motivation for studying geodesics in $\mathbb{P}_p^\mu(Y \times U)$ is practical – in Section 6, we show how one can model geodesics in $\mathbb{P}_p^\mu(Y \times U)$ in order to obtain a conditional flow-based model whose paths are easy to integrate.

A *curve* is a continuous function $\gamma_\bullet : I \to \mathbb{P}_p^\mu(Y \times U)$ where $I = (a, b) \subseteq \mathbb{R}$ is any open interval of finite length. A curve is *absolutely continuous* if there exists $m \in L^1((a, b))$ such that

$$W_p^\mu(\gamma_s, \gamma_t) \leq \int_s^t m(\tau) \, \mathrm{d}\tau \qquad \forall a < s \leq t < b. \tag{7}$$

If $(\gamma_t)$ is an absolutely continuous curve, then its metric derivative

$$|\gamma'|(t) = \lim_{s \to t} \frac{W_p^\mu(\gamma_s, \gamma_t)}{|s - t|} \tag{8}$$

exists for almost every $t \in (a, b)$, and, moreover, we almost surely have $|\gamma'|(t) \leq m(t)$ pointwise for any $m$ satisfying Equation (7) [Ambrosio et al., 2005, Theorem 1.1.2]. A curve $(\gamma_t)$ is called a *constant speed geodesic* if for all $a < s \leq t < b$, we have $W_p^\mu(\gamma_s, \gamma_t) = |t - s|W_p^\mu(\gamma_a, \gamma_b)$. It is straightforward to show that every constant speed geodesic is absolutely continuous.

**Theorem 1** ($\mathbb{P}_p^\mu(Y \times U)$ is a Geodesic Space)
*For any $\eta, \nu \in \mathbb{P}_p^\mu(Y \times U)$, there exists a constant speed geodesic between $\eta$ and $\nu$.*

When an optimal triangular coupling $\gamma^\star \in \Pi_Y(\eta, \nu)$ is induced by an injective triangular map $T^\star$, we may recover a constant speed geodesic in $\mathbb{P}_p^\mu(Y \times U)$, generalizing the McCann interpolant [McCann, 1997] to the conditional setting. We refer to Proposition 4 for sufficient conditions on $\eta, \nu$ under which such a $T^\star$ exists. Informally, samples from $(y_0, u_0) \sim \eta$ flow in a straight path at a constant speed to their destination $T^\star(y_0, u_0)$.

**Theorem 2** (Conditional McCann Interpolants)
*Fix $\eta, \nu \in \mathbb{P}_p^\mu(Y \times U)$. Suppose $T^\star(y, u) = (y, T_{\mathcal{U}}^\star(y, u))$ is an injective triangular map solving the conditional Monge problem* (1)*. Define the maps $T_t : Y \times U \to Y \times U$ for $0 \leq t \leq 1$ via $T_t = (1 - t)I + tT^\star$, and define the curve of measures $\gamma_t = [T_t]_\# \eta \in \mathbb{P}_p^\gamma(Y \times U)$. Then,*

    *(a) $(\gamma_t)$ is absolutely continuous and a constant speed geodesic between $\eta, \nu$*

    *(b) The vector field $v_t(T_t^\star(y, u)) = (0, T_U^\star(y, u) - u)$ generates the path $\gamma_t$, in the sense that $(\gamma_t, v_t)$ solve the continuity equation* (9)*.*

## 5 Conditional Benamou-Brenier Theorem

In this section, we prove a characterization of the absolutely continuous curves in $\mathbb{P}_p^\mu(Y \times U)$. As a corollary, we obtain a conditional generalization of the Benamou-Brenier Theorem [Benamou and Brenier, 2000], giving us a dynamic characterization of the conditional Wasserstein distance. Roughly speaking, all such curves are generated by a vector field on $Y \times U$ which has zero velocity in the $Y$ component. This is natural, as all measures in $\mathbb{P}_p^\mu(Y \times U)$ have a fixed $Y$-marginal $\mu$. Such a vector field can be informally seen as tangent to a curve of measures, and is the dynamic analogue of the triangular maps discussed in Section 3. More formally, given an open interval $I \subseteq \mathbb{R}$, a time-dependent Borel vector field $v : I \times Y \times U \to Y \times U$ is said to be *triangular* if there exists a Borel vector field $v^U : I \times Y \times U \to U$ such that $v_t(y, u) = (0, v_t^U(y, u))$.

**Continuity Equation.** We introduce some necessary background which allows us to link vector fields to curves of measures. The *continuity equation* $\partial_t \gamma_t + \mathrm{div}(v_t \gamma_t) = 0$ describes the evolution of a measure $\gamma_t$ which flows along a given vector field $v_t$ [Ambrosio et al., 2005, Chapter 8]. This equation must be understood in the sense of distributions, i.e. for every $\varphi$ in a space of test functions,

$$\int_I \int_{Y \times U} (\partial_t \varphi(y, u, t) + \langle v_t(y, u), \nabla_{y,u} \varphi(y, u, t) \rangle) \, \mathrm{d}\gamma_t(y, u) \, \mathrm{d}t = 0. \tag{9}$$

We consider cylindrical test functions $\varphi \in \text{Cyl}(Y \times U \times I)$, i.e. of the form $\varphi(y, u, t) = \psi(\pi^d(y, u), t)$ where $\pi^d : Y \times U \to \mathbb{R}^d$ maps $(y, u) \mapsto (\langle (y, u), e_1 \rangle, \ldots, \langle (y, u), e_d \rangle)$ and $\{e_1, e_2, \ldots, e_d\}$ is any orthonormal family in $Y \times U$. In the finite dimensional setting, one may take $\varphi \in C_c^\infty(Y \times U)$ to be smooth and compactly supported [Ambrosio et al., 2005, Remark 8.1.1].

In Appendix E, we prove Lemma 1, which is key in proving Theorem 4 below. Informally, Lemma 1 states that if the weak continuity equation (9) is satisfied for a joint distribution and triangular vector field, then the continuity equation is also satisfied for the corresponding conditional distributions and $U$ components of the vector field.

**Lemma 1** (Triangular Vector Fields Preserve Conditionals)
*Suppose $v_t(y, u) = (0, v_t^U(y, u))$ is triangular and that $(\gamma_t) \subset \mathbb{P}_p^\mu(Y \times U)$ is a path of measures such that $(v_t, \gamma_t)$ satisfy the continuity equation in the sense of distributions. Then, it follows that for $\mu$-almost every $y \in Y$, we have $\partial_t \gamma_t^y + \nabla \cdot (v_t^U(y, -)\gamma_t^y) = 0$.*

We note that having $v_t$ be triangular is sufficient, but certainly not necessary, for the conditional continuity equation to almost surely hold. For instance, the vector field in $\mathbb{R}^d$ that rotates $\mathcal{N}(0, I)$ about the origin is not triangular yet preserves all conditional distributions.

**Absolutely Continuous Curves.** In this section, we state our characterization of absolutely continuous curves in $\mathbb{P}_p^\mu(Y \times U)$. Informally, given such a curve, Theorem 3 provides us with a triangular vector field which generates the curve, in the sense that the pair solve the continuity equation.

**Theorem 3** (Absolutely Continuous Curves in $\mathbb{P}_p^\mu(Y \times U)$)
*Let $I \subset \mathbb{R}$ be an open interval, and suppose $\gamma_t : I \to \mathbb{P}_p^\mu(Y \times U)$ is an absolutely continuous in the $W_p^\mu$ metric with $|\gamma'|(t) \in L^1(I)$. Then, there exists a Borel vector field $v_t(y, u)$ such that*

*(a) $v_t$ is triangular*

*(b) $v_t \in L^p(\gamma_t, Y \times U)$ and $\|v_t\|_{L^p(\gamma_t, Y \times U)} \leq |\gamma'|(t)$ for a.e. $t$*

*(c) $(v_t, \gamma_t)$ solve the continuity equation in the sense of distributions.*

Conversely, we show in Theorem 4 that if the pair $(\gamma_t, v_t)$ solve the continuity equation and $v_t$ is triangular, then the curve $(\gamma_t)$ is absolutely continuous and $|\gamma'|(t) \leq \|v_t\|_{L^p(\gamma_t, Y \times U)}$. The main technique of this result is to study the collection of *conditional* continuity equations (which is feasible by Lemma 1) and to apply the converse of Ambrosio et al. [2005, Theorem 8.3.1]. In this setting, the infinite-dimensional result is obtained via a finite-dimensional approximation argument.

**Theorem 4** (Continuous Curves Generated by Triangular Vector Fields)
*Suppose that $\gamma_t : I \to \mathbb{P}_p^\mu(Y \times U)$ is narrowly continuous and $(v_t)$ is a triangular vector field such that $(\gamma_t, v_t)$ solve the continuity equation with $\|v_t\|_{L^p(\gamma_t, Y \times U)} \in L^1(I)$. Then, $\gamma_t : I \to \mathbb{P}_p^\mu(Y \times U)$ is absolutely continuous in the $W_p^\mu$ metric and $|\gamma'|(t) \leq \|v_t\|_{L^p(\mu, Y \times U)}$ for almost every $t$.*

As a corollary of Theorem 3 and Theorem 4, we obtain a conditional version of the Benamou-Brenier theorem [Benamou and Brenier, 2000]. Once we have our characterization of absolutely continuous curves provided by these theorems, the proof of Theorem 5 largely follows the unconditional case (see e.g. Ambrosio et al. [2005, Chapter 8]), but we include it for the sake of completeness.

**Theorem 5** (Conditional Benamou-Brenier)
*Let $1 < p < \infty$. For any $\eta, \nu \in \mathbb{P}_p^\mu(Y \times U)$, we have*

$$W_p^{p,\mu}(\eta, \nu) = \min_{(\gamma_t, v_t)} \left\{ \int_0^1 \|v_t\|_{L^p(\mu_t)}^p \, \mathrm{d}t \mid (v_t, \gamma_t) \text{ solve (9)}, \gamma_0 = \eta, \gamma_1 = \nu, \text{ and } v_t \text{ is triangular} \right\}.$$

# 6 COT Flow Matching

We have thus far seen that the COT problem (3) admits a dynamic formulation by Theorem 5, where one may take the underlying vector fields to be triangular. We use these results to design a principled model for conditional generation based on flow matching [Lipman et al., 2022, Albergo et al., 2023b, Liu et al., 2022, Tong et al., 2023, Pooladian et al., 2023]. We hereafter use the squared-distance cost (i.e. $p = 2$).

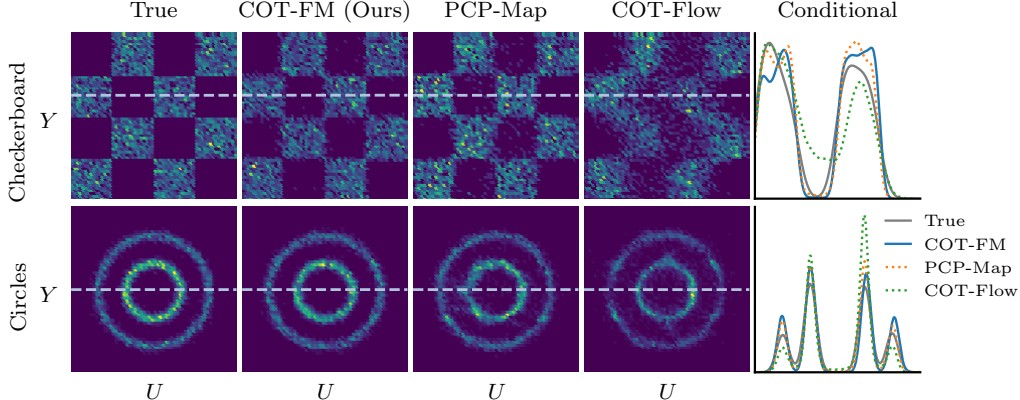

Figure 1: Samples from the ground-truth joint target distribution and the various models. Samples from COT-FM more closely match the ground-truth distribution than the baselines. In the final column, we plot conditional KDEs for samples drawn conditioned on the $y$ value indicated by the dashed horizontal line. See Appendix F for a larger figure and additional results.

**Flow Matching.** We assume that we have access to samples $z_0 = (y_0, u_0) \sim \eta(y_0, u_0) \in \mathbb{P}_p^\mu(Y \times U)$ from a source measure, and samples $z_1 = (y_1, u_1) \sim \nu(y_1, u_1) \in \mathbb{P}_p^\mu(Y \times U)$ from a target measure. Let $z = (z_0, z_1) \sim \rho(z_0, z_1) \in \Pi(\eta, \nu)$ be any coupling of the source and target measure. We specify a collection of measures and vector fields on $Y \times U$ via

$$\gamma_t(y, u \mid z) = \mathcal{N}(y, u \mid tz_1 + (1 - t)z_0, C) \qquad v_t(y, u \mid z) = z_1 - z_0 \qquad (10)$$

where $C$ is any trace-class covariance operator [Da Prato and Zabczyk, 2014]. As is standard in flow matching [Lipman et al., 2022, Kerrigan et al., 2024], we obtain from Equations (10) a marginal measure $\gamma_t(y, u)$ and vector field $v_t(y, u)$ satisfying the continuity equation via

$$\gamma_t(y, u) = \int_{(Y \times U)^2} \gamma_t(y, u \mid z)\, \mathrm{d}\rho(z) \qquad v_t(y, u) = \int_{(Y \times U)^2} v_t(y, u \mid z) \frac{\mathrm{d}\gamma_t(y, u \mid z)}{\mathrm{d}\gamma_t(y, u)}\, \mathrm{d}\rho(z).$$
$$(11)$$

This marginal path $(\gamma_t)_{t=0}^1$ interpolates between the source measure ($t = 0$) and a smoothed version of the target measure ($t = 1$). To transform source samples from $\eta$ into target samples from $\nu$, we seek to learn the intractable vector field $v_t(y, u)$ with a model $v^\theta(t, y, u)$ by minimizing the loss[1]

$$\mathcal{L}(\theta) = \mathbb{E}_{t, \rho(z), \gamma_t(y, u \mid z)} \left\| v^\theta(t, y, u) - v_t(y, u \mid z) \right\|^2 \qquad (12)$$

which has the same $\theta$-gradient as the MSE loss to the true vector field $u_t(y, u)$ [Tong et al., 2023].

**COT Flow Matching.** In the preceding section, $\rho(z)$ may be an arbitrary coupling between $\eta$ and $\nu$. Motivated by Proposition 1, we will choose $\rho$ to be a COT coupling. Under sufficient regularity conditions (see Appendix B), this COT plan will be induced by a triangular map. In turn, Theorem 2 gives us that this triangular map is generated by a triangular vector field of the form (10). Thus, we parametrize our model $u^\theta$ to also be triangular. Moreover, we recover the optimal dynamic transport given in Theorem 5 as $\mathrm{Tr}(C) \to 0$ by a pointwise application of [Tong et al., 2023, Proposition 3.4].

Given a collection of samples $\{z_0^i, z_1^i\}_{i=1}^n$ drawn from $\eta$ and $\nu$, we approximate a conditional optimal coupling $\rho$ using standard numerical techniques with the cost function $c_\epsilon(y_0, u_0, y_1, u_1) = |y_1 - y_0|^2 + \epsilon |u_1 - u_0|^2$ for some $0 < \epsilon \ll 1$. Intuitively, such a cost penalizes mass transfer along the $Y$ dimension, which is precisely the constraint sought in the COT problem (3). As $\epsilon \downarrow 0$, we recover the true optimal triangular map [Carlier et al., 2010, Hosseini et al., 2023]. The COT coupling

---

[1]Previous work has referred to this as the *conditional flow matching loss* [Tong et al., 2023], which is not to be confused with the notion of conditioning that we focus on in this work.

Table 1: Distances between the ground-truth and generated joint distributions for the 2D datasets. Our method (COT-FM) obtains lower distances than the considered baselines. Average results $\pm$ one standard deviation are reported across five test sets, with the lowest average distance in bold.

| | Checkerboard | | Moons | | Circles | | Swissroll | |
|---|---|---|---|---|---|---|---|---|
| | $W_2$ (1e-2) | MMD (1e-3) | $W_2$ (1e-2) | MMD (1e-3) | $W_2$ (1e-2) | MMD (1e-3) | $W_2$ (1e-2) | MMD (1e-3) |
| PCP-Map | $6.27_{\pm0.81}$ | $0.21_{\pm0.13}$ | $8.44_{\pm1.09}$ | $0.22_{\pm0.10}$ | $6.19_{\pm0.43}$ | $\mathbf{0.20}_{\pm0.17}$ | $5.35_{\pm0.93}$ | $0.16_{\pm0.13}$ |
| COT-Flow | $8.20_{\pm0.49}$ | $0.26_{\pm0.16}$ | $18.49_{\pm2.22}$ | $1.32_{\pm0.79}$ | $10.04_{\pm1.69}$ | $0.24_{\pm0.22}$ | $6.47_{\pm0.69}$ | $0.19_{\pm0.19}$ |
| FM | $8.81_{\pm0.58}$ | $0.24_{\pm0.20}$ | $15.55_{\pm0.77}$ | $1.85_{\pm0.22}$ | $7.03_{\pm0.17}$ | $0.45_{\pm0.11}$ | $8.18_{\pm0.34}$ | $0.58_{\pm0.09}$ |
| COT-FM (Ours) | $\mathbf{4.69}_{\pm1.00}$ | $\mathbf{0.17}_{\pm0.13}$ | $\mathbf{6.50}_{\pm1.41}$ | $\mathbf{0.13}_{\pm0.10}$ | $\mathbf{5.56}_{\pm0.43}$ | $\mathbf{0.20}_{\pm0.04}$ | $\mathbf{4.64}_{\pm1.26}$ | $\mathbf{0.15}_{\pm0.19}$ |

can either be precomputed for small datasets or computed on each minibatch drawn during training. While the use of minibatches is a computational necessity, we find that surprisingly small batch sizes still yields accurate approximations of the true COT mapping using our COT-FM method. See Appendix G.

After training, we obtain a learned triangular vector field $v^\theta(t, y, u)$. Given an arbitrary fixed $y \in Y$, we may approximately sample from the target $\nu(u \mid y)$ by sampling $u_0 \sim \eta(u_0 \mid y)$ and numerically solving the corresponding flow equation $\partial_t(y, u_t) = v^\theta(t, y, u_t)$ with initial condition $(y, u_0)$.

**Source Measure.** Our framework is agnostic to the choice of source measure $\eta$, allowing for flexibility in the modeling process. The main requirement is that the $Y$-marginals of the source $\eta$ and target $\eta$ must match. In some scenarios, this is trivially satisfied. If one is interested in using a source distribution which is simply random noise, one may take $\eta(y_0, u_0) = \pi_\#^Y \nu(y_0) \otimes \eta_U(u_0)$ to be the product of two independent distributions where $\eta_U$ is arbitrary, e.g. Gaussian noise.

## 7 Experiments

We now illustrate our methodology (COT-FM) on a variety of conditional simulation tasks. We compare our method against several competitive baselines, namely PCP-Map [Wang et al., 2023], COT-Flow [Wang et al., 2023], and WaMGAN [Hosseini et al., 2023]. These baselines are chosen as they reflect current state-of-the-art approaches to learning COT maps. We additionally compare against flow matching [Lipman et al., 2022, Wildberger et al., 2024] without COT, i.e. where the coupling between the source and target measures is the independent coupling $\rho(z_0, z_1) = \eta \otimes \nu$. This baseline serves as an ablation for the COT component of our model.

Table 2: Statistical distances between MCMC and posterior samples $u \sim \nu(u \mid y)$ for each method on the LV dataset. Average results $\pm$ one standard deviation reported across five test sets.

| | $W_2$ (1e-2) | MMD (1e-3) |
|---|---|---|
| PCP-Map | $5.04_{\pm0.05}$ | $2.67_{\pm2.1}$ |
| COT-Flow | $4.86_{\pm1.1}$ | $\mathbf{0.83}_{\pm0.50}$ |
| FM | $11.41_{\pm0.26}$ | $2.65_{\pm0.14}$ |
| COT-FM (Ours) | $\mathbf{4.02}_{\pm0.06}$ | $0.95_{\pm0.03}$ |

Overall, our method (COT-FM) typically outperforms these baselines across the diverse and challenging set of tasks we consider. We find that PCP-Map [Wang et al., 2023] is a strong baseline, but we emphasize that this model relies on the use of an input-convex neural network [Amos et al., 2017] and it is hence unclear how to adapt this method to e.g. images. Appendix F contains further details and results for all of our experiments.[2]

**2D Synthetic Data.** We first consider synthetic distributions where $Y = U = \mathbb{R}$. Our source measure is taken to be the independent product $\eta(y, u) = \pi_\#^Y \nu \otimes \mathcal{N}(0, 1)$. We plot ground-truth joint distributions and samples for two datasets in Figure 1. See Appendix F for additional results. Samples from our method (COT-FM) closely match those from the ground-truth distribution, whereas samples from PCP-Map and COT-Flow [Wang et al., 2023] can produce samples in regions of zero support under the ground-truth distribution. In Table 1, we provide a quantitative analysis, where we measure the $W_2$ and MMD distances between the generated and ground-truth joint distributions. This is motivated by Proposition 1, as triangular maps which couple the joint distributions necessarily couple the conditional distributions. Our method outperforms the baselines across all metrics.

---

[2]Code for all of our experiments is available at `https://github.com/GavinKerrigan/cot_fm`

**Lotka-Volterra (LV) Dynamical System.** Here we estimate parameters of the Lotka-Volterra (LV) model given only noisy observations of its solution. The LV model has parameters $u = (\alpha, \beta, \gamma, \delta) \in \mathbb{R}^4_{\geq 0}$ and a pair of coupled nonlinear ODEs of the form

$$\frac{\mathrm{d}p_1(t)}{\mathrm{d}t} = \alpha p_1 - \beta p_1 p_2 \qquad \frac{\mathrm{d}p_2(t)}{\mathrm{d}t} = -\gamma p_2 + \delta p_1 p_2 \qquad (13)$$

whose solution $p(t) = (p_1(t), p_2(t)) \in \mathbb{R}^2_{\geq 0}$ represents the number of prey and predator species at time $t \in [0, T]$. Following Alfonso et al. [2023], we assume $p(0) = (30, 1)$ and that $\log(u) \sim \mathcal{N}(m, 0.5I)$ with $m = (-0.125, -3, -0.125, -3)$. Given parameters $u \in \mathbb{R}^4_{\geq 0}$, we simulate Equation (13) for $t \in \{0, 2, \ldots, 20\}$ to obtain a solution $z(u) \in \mathbb{R}^{22}_{\geq 0}$. An observation $y \in \mathbb{R}^{22}_{\geq 0}$ is obtained by the addition of log-normal noise, i.e. $\log(y) \sim \mathcal{N}(\log(z(u)), 0.1I)$. We thus may simulate many $(y, u)$ pairs from the target measure for training.

As a benchmark, we follow the settings of Alfonso et al. [2023] and choose parameters $u = (0.83, 0.041, 1.08, 0.04)$ to generate a single observation $y$ as described above. In Figure 2, we plot a histogram of $10,000$ samples from the posterior $\nu(u \mid y)$ of COT-FM.

Since the ground-truth posterior is intractable, we compare against differential evolution Metropolis MCMC [Braak, 2006]. Samples from our method qualitatively resemble those from MCMC, and the posterior mode is typically close to the true unknown $u$ (shown in red). Our method is quantitatively closest to the MCMC samples in the $W_2$ metric, and competitive in the MMD metric (Table 2).

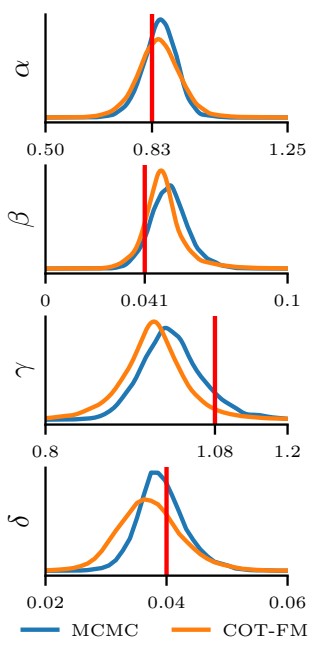

Figure 2: Sample KDEs on the Lotka-Volterra inverse problem. The red lines denote the true parameter values.

**Darcy Flow Inverse Problem.** Here we consider an infinite-dimensional Bayesian inverse problem from the 2D Darcy flow PDE. The setting is adapted from Hosseini et al. [2023]. We opt to compare against WaMGAN [Hosseini et al., 2023], as this is currently the only other extant amortized function-space COT method, and FFM [Kerrigan et al., 2023] as a function-space flow matching ablation.

Table 3: Predictive performance of the generated samples on the Darcy flow inverse problem. Average result $\pm$ one standard deviation obtained on 5 test sets of 5,000 samples each.

|  | MSE (1e-2) | CRPS (1e-2) |
|---|---|---|
| WaMGAN | 6.55±0.07 | 18.75±0.10 |
| FFM | 7.30±0.07 | **15.47**±0.06 |
| COT-FFM (Ours) | **5.40**±0.08 | 15.56±0.08 |

The Darcy flow PDE is an elliptic equation on a smooth domain $\Omega \subseteq \mathbb{R}^d$ which relates a permeability field $\exp(u)$, a pressure field $\rho$, and a source term $f$ via $-\mathrm{div}\,\exp(u)\nabla\rho = f$ on $\Omega$ subject to $\rho = 0$ on $\partial\Omega$. Our goal is to recover the permeability $u$ from noisy measurements $y$ of the pressure $\rho$. Both the unknown $u$ and observations $y$ are functions and thus infinite-dimensional. To define our target measure, we specify a prior $\nu(u) = \mathcal{N}(0, C)$ with a Matérn kernel $C$ of lengthscale $\ell = 1/2$ and $\nu = 3/2$. Given $u \sim \eta(u)$, the Darcy flow PDE is solved numerically [Alnæs et al., 2015] to obtain a solution $\rho(u)$ observed at some finite but arbitrary number of points $\{x_1, \ldots, x_n\} \subset \mathbb{R}^2$. An observation $y(u)$ is obtained by adding Gaussian noise to each observation, i.e. $y(u) \sim \mathcal{N}(\rho(u), \sigma^2 I)$ where $\sigma = 2.5 \times 10^{-2}$. We implement all models via a Fourier Neural Operator [Li et al., 2020], allowing us to work with arbitrary discretizations, as required by the functional nature of this problem.

We provide an illustration in Figure 3. As the true posterior is intractable, we compare against preconditioned Crank-Nicolson (pCN) [Cotter et al., 2013], a function-space MCMC method. In Figure 3, we plot the mean posteriors obtained from the various methods. Qualitatively, both COT-FM and FFM are good approximations to pCN, while WaMGAN has visual artifacts. However, the MSE

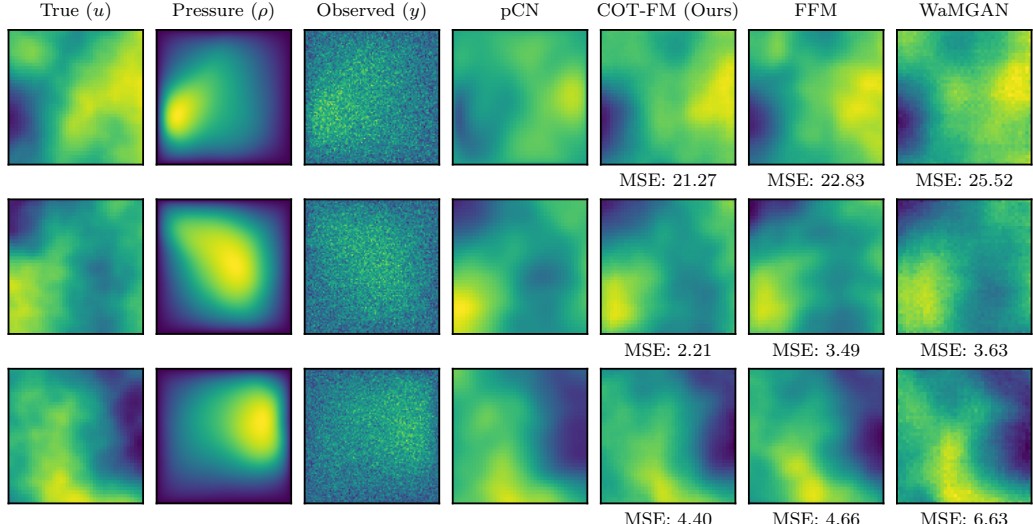

Figure 3: Darcy flow illustration. Several true permeability fields $u$ are shown, as well as the pressure field $\rho$ and its observed, noisy version $y$. We compare an ensemble average of posterior samples from the various methods against MCMC (pCN) [Cotter et al., 2013]. COT-FM achieves the lowest MSE to pCN. We note here that WaMGAN has clear visual artifacts despite achieving reasonable MSE and CRPS scores.

between our method and the pCN mean is lower than that of FFM. Table 3 provides a quantitative comparison between the methods on a test set of 5,000 samples, where we measure MSE and CRPS [Hersbach, 2000]. We compare the ensemble mean of 10 samples against the true $u$ value as running pCN for each observation is prohibitively expensive. COT-FM outperforms FFM and WaMGAN in terms of MSE and is on-par with FFM in terms of CRPS. See Appendix F for further details.

## 8    Conclusion

We analyze conditional optimal transport from a geometric and dynamical point of view. Our analysis culminates in the characterization of absolutely continuous curves of measures in a conditional Wasserstein space, resulting in a conditional analog of the Benamou-Brenier Theorem.

We use these result to build on the framework of triangular transport and flow matching to develop simulation-free methods for conditional generative models. Our methods are applicable across a wide class of problems, and we demonstrate our methodology on several challenging inverse problems.

**Limitations and Broader Impacts.**    A limitation COT-FM is that computing the full COT plan can be expensive for large datasets, necessitating the use of minibatch approximations potentially resulting in sub-optimal plans. While this approximation does not limit the practical applicability of our method, an interesting challenge is to characterize the precise relationship between this minibatch approximation and the full COT plan. Moreover, computing the COT plan incurs a small additional computational cost compared to standard flow matching. As with all generative models, a potential negative impact is the potential for disinformation through generated samples being purported as real.

## Acknowledgments and Disclosure of Funding

This research was supported by the Hasso Plattner Institute (HPI) Research Center in Machine Learning and Data Science at the University of California, Irvine, by the National Science Foundation under award 1900644, and by the National Institutes of Health under award R01-LM013344.

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

## Appendix: Table of Contents

## A  Optimal Transport

We provide here a brief and informal overview of optimal transport in the standard unconditional setting. For more details, we refer to the standard references of Villani et al. [2009], Santambrogio [2015], Ambrosio et al. [2005]. Let $X$ be a separable metric space and fix a cost function $c : X \times X \to \mathbb{R} \cup \{+\infty\}$. Suppose we have two Borel measures $\eta, \nu \in \mathbb{P}(X)$. The *Monge problem* seeks to find a measurable transport map $T : X \to X$ minimizing the expected cost of transport, i.e. corresponding to the optimization problem

$$\inf_T \left\{ \int_X c(x, T(x)) \, \mathrm{d}\eta(x) \mid T_\# \eta = \nu \right\}. \tag{14}$$

This optimization problem is challenging, though, as it involves a nonlinear constraint and the set of feasible maps may be empty. In contrast, the *Kantorovich problem* is a relaxation which seeks to find an optimal coupling $\gamma \in \Pi(\eta, \nu)$, i.e. a probability distribution over $X \times X$ with marginals $\eta, \nu$, which solves

$$\inf_\gamma \left\{ \int_{X \times X} c(x_0, x_1) \, \mathrm{d}\gamma(x_0, x_1) \mid \gamma \in \Pi(\eta, \nu) \right\}. \tag{15}$$

Under fairly weak conditions (e.g., the cost is lower semicontinuous and bounded from below [Ambrosio et al., 2013, Theorem 2.5]), minimizers to the Kantorovich problem are guaranteed to exist. If the cost function is $c(x_0, x_1) = |x_0 - x_1|^p$ for some $1 < p < \infty$, under sufficient regularity conditions on $\eta$ a solution $T^\star$ to the Monge problem is guaranteed to exist and, moreover, the coupling $\gamma^\star = (I, T^\star)_\# \eta$ is optimal for the Kantorovich problem. See Ambrosio et al. [2013, Chapter 2] and Ambrosio et al. [2005, Theorem 6.2.10].

**Wasserstein Space.** In the special case that $c(x_0, x_1) = |x_0 - x_1|^p$ for $1 \leq p < \infty$, and $\eta, \nu \in \mathbb{P}_p(X)$, the Kantorovich problem admits a finite-cost solution. The cost of such an optimal coupling is the $p$-*Wasserstein distance*

$$W_p^p(\eta, \nu) = \min_\gamma \left\{ \int_{X \times X} |x_0 - x_1|^p \, \mathrm{d}\gamma(x_0, x_1) \mid \gamma \in \Pi(\eta, \nu) \right\} \tag{16}$$

which, as the name suggests, is a metric on the space $\mathbb{P}_p(X)$ [Ambrosio et al., 2005, Section 7.1] [Santambrogio, 2015, Section 5.1]. The Wasserstein distance admits a *dynamical* formulation via the Benamou-Brenier theorem [Benamou and Brenier, 2000]. Namely, the $p$-Wasserstein distance can be obtained by finding a time-dependent vector field transforming $\eta$ to $\nu$ across time $t \in [0, 1]$ with minimal energy:

$$W_p(\eta, \nu) = \min_{(\gamma_t, v_t)} \left\{ \int_0^1 \int_X |v_t(x)|^p \, \mathrm{d}\gamma_t(x) \, \mathrm{d}t \mid \gamma_0 = \eta, \gamma_1 = \nu, \partial_t \gamma_t + \mathrm{div}(v_t \gamma_t) = 0 \right\}. \tag{17}$$

Here, we constrain our minimization problem over the set of measures and vector fields $(\gamma_t, v_t)$ interpolating between $\eta$ and $\nu$, satisfying a continuity equation (see Section 5). In Section 4, we study a generalization of the Wasserstein distances for conditional optimal transport problems. In particular, Theorem 5 provides a generalization of the Benamou-Brenier theorem to the conditional setting which recovers a conditional Wasserstein distance.

## B Conditional Optimal Transport

This section contains additional discussion regarding the static COT problem, supplementing Section 3.1. We refer to Hosseini et al. [2023], Baptista et al. [2020], and Chemseddine et al. [2023] for further results and details.

Given a source and target measures $\eta, \nu \in \mathbb{P}_p^\mu(Y \times U)$, the *conditional Monge problem* seeks to find a triangular mapping solving

$$\inf_T \left\{ \int_{Y \times U} c(y, u, T(y, u)) \, \mathrm{d}\eta(y, u) \mid T_\# \eta = \nu, T : (y, u) \mapsto (y, T_U(y, u)) \right\}. \tag{18}$$

The conditional Monge problem also admits a relaxation under which one only considers couplings whose $Y$-components are almost surely equal. To that end, we consider the subset $\mathscr{C} \subset (Y \times U)^2$ whose $Y$ components are identical, i.e.,

$$\mathscr{C} := \left\{ (y_0, u_0, y_1, u_1) \in (Y \times U)^2 \mid y_0 = y_1 \right\} \tag{19}$$

and we define the set of $(Y)$-*restricted probability measures* $\mathcal{R}_Y \subset \mathbb{P}\left((Y \times U)^2\right)$ such that every $\gamma \in \mathcal{R}_Y$ is concentrated on $\mathscr{C}$. In other words, if $\gamma \in \mathcal{R}_Y$, then samples $(y_0, u_0, y_1, u_1) \sim \gamma$ have $y_0 = y_1$ almost surely. In addition, for any $\eta, \nu \in \mathbb{P}(Y \times U)$, we define the set of *triangular couplings* $\Pi_Y(\eta, \nu)$ to be the probability measures in $\mathcal{R}_Y$ whose marginals are $\eta$ and $\nu$, i.e.

$$\Pi_Y(\eta, \nu) = \left\{ \gamma \in \mathcal{R}_Y \mid \pi_\#^{1,2} \gamma = \eta, \pi_\#^{3,4} \gamma = \nu \right\}. \tag{20}$$

The *conditional Kantorovich problem* seeks a triangular coupling $\gamma^\star$ solving

$$\inf_\gamma \left\{ \int_{(Y \times U)^2} c(y_0, u_0, y_1, u_1) \, \mathrm{d}\gamma(y_0, u_0, y_1, u_1) \mid \gamma \in \Pi_Y(\eta, \nu) \right\}. \tag{21}$$

Hosseini et al. [2023] prove the existence of minimizers to the conditional Kantorovich and Monge problems under very general assumptions. Moreover, optimal couplings to the conditional Kantorovich problem induce optimal couplings for $\mu$-almost every conditional measure. Assuming sufficient regularity assumptions on the conditional measures, unique solutions to the conditional Monge problem exist. We restate these results here for the sake of completeness.

**Proposition 3** (Prop 3.3 [Hosseini et al., 2023])
*Fix $\eta, \nu \in \mathbb{P}^\mu(Y \times U)$. Suppose the cost function $c$ is continuous, $\inf c > -\infty$, and there exists a finite cost coupling $\gamma \in \Pi_Y(\eta, \nu)$. Then, the conditional Kantorovich problem admits a minimizer $\gamma^\star$. Moreover, $\gamma^{\star, y_0}(y_1, u_0, u_1) = \hat{\gamma}^{\star, y_0}(u_0, u_1)\delta(y_1 - y_0)$ where for $\mu$-almost every $y$ the measure $\gamma^{\star, y}$ is an optimal coupling for $\eta^y, \nu^y$ under the cost $c^y(u_0, u_1) = c(y, u_0, y, u_1)$*

**Proposition 4** (Prop 3.8 [Hosseini et al., 2023])
*Fix $1 < p < \infty$ and $\eta, \nu \in \mathbb{P}_p^\mu(Y \times U)$. Suppose $c(y_0, u_0, y_1, u_1) = |u_0 - u_1|^p$. If $\eta^y$ assign zero measure to Gaussian null sets for $\mu$-almost every $y$, then there is a unique solution $T^\star$ to the conditional Monge problem, and $\gamma^\star = (I, T^\star)_{\#}\eta$ is the unique solution to the conditional Kantorovich problem. If $\nu^y$ also assign zero measure to Gaussian null sets for $\mu$-almost every $y$, then $T^\star$ is injective $\eta$-almost everywhere.*

## C   Closed-Form Conditional Wasserstein Distance for Gaussian Measures

In this section, we provide additional details and results regarding the closed-form conditional Wasserstein distance for Gaussian distributions. See Section 4 in the main paper.

Suppose $Y = \mathbb{R}^d$ and $U = \mathbb{R}^{d'}$ are Euclidean spaces (of possibly different dimensions), and that $\eta, \nu \in \mathbb{P}_p^\mu(Y \times U)$ are Gaussians of the form

$$\eta = \mathcal{N}\left(\begin{bmatrix} m \\ m_u^\eta \end{bmatrix}, \begin{bmatrix} \Sigma & \Lambda^\eta \\ \Lambda^{\eta^T} & \Sigma_u^\eta \end{bmatrix}\right) \qquad \nu = \mathcal{N}\left(\begin{bmatrix} m \\ m_u^\nu \end{bmatrix}, \begin{bmatrix} \Sigma & \Lambda^\nu \\ \Lambda^{\nu^T} & \Sigma_u^\nu \end{bmatrix}\right) \tag{22}$$

where $m \in \mathbb{R}^d$, $m_u^\eta, m_u^\nu \in \mathbb{R}^{d'}$, and the (block) covariance matrices are $\Sigma \in \mathbb{R}^{d \times d}$, $\Lambda^\eta, \Lambda^\nu \in \mathbb{R}^{d \times d'}$, and $\Sigma_u^\eta, \Sigma_u^\nu \in \mathbb{R}^{d' \times d'}$.

This form is chosen to ensure that $\eta$ and $\nu$ have equal $Y$-marginals. It follows that $\mu = \pi_{\#}^Y \eta = \pi_{\#}^Y \nu = \mathcal{N}(m, \Sigma)$. Let

$$Q^\eta = \Sigma_u^\eta - \Lambda^{\eta^T} \Sigma^{-1} \Lambda^\eta \qquad Q^\nu = \Sigma_u^\nu - \Lambda^{\nu^T} \Sigma^{-1} \Lambda^\nu \qquad R = (\Lambda^\eta - \Lambda^\nu)^T \Sigma^{-1}. \tag{23}$$

We have that the conditionals $\eta^y, \nu^y$ are available in closed-form:

$$\eta^y = \mathcal{N}\left(m_u^\eta + \Lambda^{\eta^T} \Sigma^{-1}(y - m), Q^\eta\right) \qquad \nu^y = \mathcal{N}\left(m_u^\nu + \Lambda^{\nu^T} \Sigma^{-1}(y - m), Q^\nu\right). \tag{24}$$

Thus, for any fixed $y$, we use the known closed-form unconditional Wasserstein distance to obtain

$$W_2^2(\eta^y, \nu^y) = \left|m_u^\eta - m_u^\nu + R(y - m)\right|^2 + \mathrm{Tr}\left(Q^\eta + Q^\nu - 2\left((Q^\eta)^{1/2} Q^\nu (Q^\eta)^{1/2}\right)^{1/2}\right). \tag{25}$$

We now take an expectation over $y \sim \mu = \mathcal{N}(m, \Sigma)$ to compute $W_2^{\mu, 2}$. Observe that $R(y - m) \sim \mathcal{N}(0, R\Sigma R^\mathsf{T})$ and that $\mathbb{E}_{y \sim \mu}[|R(y - m)|^2] = \mathrm{Tr}(R\Sigma R^\mathsf{T})$. Thus,

$$W_2^{\mu, 2}(\eta, \nu) = \mathbb{E}_{y \sim \mu}\left[W_2^2(\eta^y, \nu^y)\right] \tag{26}$$

$$= \mathbb{E}_{y \sim \mu}\left[|m_u^\eta - m_u^\nu|^2 + 2\langle m_u^\eta - m_u^\nu, R(y - m)\rangle + |R(y - m)|^2\right] \tag{27}$$

$$+ \mathrm{Tr}\left(Q^\eta + Q^\nu - 2\left((Q^\eta)^{1/2} Q^\nu (Q^\eta)^{1/2}\right)^{1/2}\right)$$

$$= |m_u^\eta - m_u^\nu|^2 + \mathrm{Tr}\left(Q^\eta + Q^\nu - 2\left((Q^\eta)^{1/2} Q^\nu (Q^\eta)^{1/2}\right)^{1/2} + R\Sigma R^\mathsf{T}\right). \tag{28}$$

This form, perhaps unsurprisingly, closely resembles the unconditional Wasserstein distance between two Gaussians, except for the presence of an additional $\mathrm{Tr}(R\Sigma R^\mathsf{T})$ term. Note that when $\eta, \nu$ have uncorrelated $Y, U$ components, we precisely recover $W_2^2(\pi_{\#}^U \eta, \pi_{\#}^U \nu)$ as one may expect. As a special case of interest, if $Y = U = \mathbb{R}$ and

$$\eta = \mathcal{N}(0, I) \qquad \nu = \mathcal{N}\left(0, \begin{bmatrix} 1 & \rho \\ \rho & 1 \end{bmatrix}\right) \qquad |\rho| < 1 \tag{29}$$

then we obtain as a special case of Equation (26) that $W_2^{\mu,2}(\eta, \nu) = 2(1 - \sqrt{1 - \rho^2})$. This is zero if and only if $\rho = 0$, i.e. $\eta = \nu$.

# D   Proofs: Section 4

In this section, we provide detailed proofs of our claims in Section 4, regarding the metric properties of the conditional Wasserstein space.

## D.1   Metric Properties

We first note that $W_p^\mu(\eta, \nu)$ may be viewed as the minimal value of the constrained Kantorovich problem in Equation (3) when one takes the cost to be the metric on the space $Y \times U$. Similar results, relating the conditional Wasserstein distance to triangular couplings, have appeared previously, but our proof is independent of these prior works [Chemseddine et al., 2023, Gigli, 2008].

**Proposition 5** (Equivalent Formulation of the Conditional Wasserstein Distance)
*Fix $\eta, \nu \in \mathbb{P}_p^\mu(Y \times U)$ and $1 \le p < \infty$. Then, $W_p^\mu(\eta, \nu)$ is well-defined, finite, and*

$$W_p^{\mu,p}(\eta, \nu) = \min_\gamma \left\{ \int_{(Y \times U)^2} d^p(y_0, u_0, y_1, u_1) \, \mathrm{d}\gamma \mid \gamma \in \Pi_Y(\eta, \nu) \right\} \tag{30}$$

*where $W_p^{\mu,p}(\eta, \nu)$ represents the p-th power of the conditional p-Wasserstein distance.*

*Proof.* The cost function $d^p$ is clearly continuous and non-negative, and hence by Proposition 3 it suffices to exhibit a finite-cost coupling $\gamma \in \Pi_Y(\eta, \nu)$ between $\eta$ and $\nu$. Indeed, take the conditionally independent coupling

$$\gamma(y_0, u_0, y_1, u_1) = \eta(u_0 \mid y_1)\nu(u_1 \mid y_1)\delta(y_1 - y_0)\mu(y_1) \tag{31}$$

which is clearly in $\Pi_Y(\eta, \nu)$. We then have that

$$\int_{(Y \times U)^2} d^p(y_0, u_0, y_1, u_1) \, \mathrm{d}\gamma(y_0, u_0, y_1, u_1) = \int_{(Y \times U)^2} \|(y_0, u_0) - (y_1, u_1)\|_{Y \times U}^p \, \mathrm{d}\gamma(y_0, u_0, y_1, u_1)$$

$$\le 2^p \int_{(Y \times U)^2} \left( \|(y_0, u_0)\|_{Y \times U}^p + \|(y_1, u_1)\|_{Y \times U}^p \right) \, \mathrm{d}\gamma(y_0, u_0, y_1, u_1)$$

$$= 2^p \left( \int_{Y \times U} \|(y_0, u_0)\|_{Y \times U}^p \, \mathrm{d}\eta(y_0, u_0) + \int_{Y \times U} \|(y_1, u_1)\|_{Y \times U}^p \, \mathrm{d}\nu(y_1, u_1) \right) < +\infty.$$

Hence, Equation (30) admits a minimizer $\gamma^\star \in \Pi_Y(\eta, \nu)$. By Proposition 3, this minimizer may be taken to have the form $\gamma^\star = \gamma^{\star,y_1}(u_0, u_1)\delta(y_1 - y_0)\mu(y_1)$ where $\gamma^{\star,y_1}(u_0, u_1)$ is $\mu(y_1)$-almost surely an optimal coupling between $\eta^{y_1}, \nu^{y_1}$ for the cost $|u_1 - u_0|^p$. Thus,

$$\int_{(Y \times U)^2} d^p \, \mathrm{d}\gamma^\star = \int_Y \int_{U^2} |u_1 - u_0|^p \, \mathrm{d}\gamma^{\star,y}(u_0, u_1) \, \mathrm{d}\mu(y) \tag{32}$$

$$= \int_Y W_p^p(\eta^y, \nu^y) \, \mathrm{d}\mu(y) = W_p^{p,\mu}(\eta, \nu). \tag{33}$$

Here, we emphasize that the $\mu$-almost sure uniqueness of the disintegrations of $\eta, \nu$ along $Y$ result in a well-defined expression.

Moreover, if $\eta \in \mathbb{P}_p^\mu(Y \times U)$ it follows that $\eta^y \in \mathbb{P}_p(U)$ for $\mu$-a.e. $y$, because

$$\int_Y \int_U |u|^p \, \mathrm{d}\eta^y(u) \, \mathrm{d}\mu(y) \le \int_Y \int_U |(y, u)|^p \, \mathrm{d}\eta^y(u) \, \mathrm{d}\mu(y) \tag{34}$$

$$= \int_{Y \times U} |(y, u)|^p \, \mathrm{d}\eta(y, u) < +\infty. \tag{35}$$

Thus all considered $p$-Wasserstein distances on $U$ are finite. $\qquad\square$

We now proceed to prove several metric properties of our distance.

**Proposition 2** (Some Properties of $W_p^\mu$)
*Let $1 \le p < \infty$.*

(a) $W_p^\mu$ *is well-defined, finite, and equals the minimal conditional Kantorovich cost.*

(b) $W_p^\mu$ *is a metric on the space $\mathbb{P}_p^\mu(Y \times U)$.*

(c) *There does not exist $C > 0$ such that $W_p^\mu(\eta, \nu) \le C W_p(\eta, \nu)$ for all $\eta, \nu \in \mathbb{P}_p^\mu(Y \times U)$.*

(d) *For all $\eta, \nu \in \mathbb{P}_p^\mu(Y \times U)$, $W_p\left(\pi_\#^U \eta, \pi_\#^U \nu\right) \le W_p^\mu(\eta, \nu)$ and $W_p(\eta, \nu) \le W_p^\mu(\eta, \nu)$.*

*Proof.* **Part (a).** This is simply a restatement of Proposition 5.

**Part (b).** Fix $\eta, \nu, \rho \in \mathbb{P}_p^\mu(Y \times U)$. Since $W_p$ is a metric on $\mathbb{P}_p(U)$, we immediately obtain the symmetry of $W_p^\mu$. Moreover, we have that $W_p^\mu(\eta, \nu) = 0$ if and only if $\eta^y = \nu^y$ for $\mu$-almost every $y$. Thus, if $W_p^\mu(\eta, \nu) = 0$ and $E \subseteq Y \times U$ is Borel measurable,

$$\eta(E) = \int_Y \eta^y(E^y) \, d\mu(y) = \int_Y \nu^y(E^y) \, d\mu(y) = \nu(E). \tag{36}$$

which shows that $\eta = \nu$. Here, $E^y = \{u \mid (y, u) \in E\}$ is the $y$-slice of $E$. Conversely, if $\eta = \nu$, then $\eta^y = \nu^y$ up to a $\mu$-null set by the essential uniqueness of disintegrations. Thus, $W_p^\mu(\eta, \nu) = 0$ if and only if $\eta = \nu$.

By Minkowski's inequality and the triangle inequality for $W_p$ on $\mathbb{P}_p(U)$, we see

$$W_p^\mu(\eta, \nu) \le \left(\mathbb{E}_{y \sim \mu}\left[(W_p(\eta^y, \rho^y) + W_p(\rho^y, \nu^y))^p\right]\right)^{1/p} \tag{37}$$

$$\le \mathbb{E}_{y \sim \mu}[W_p^p(\eta^y, \rho^y)]^{1/p} + \mathbb{E}_{y \sim \mu}[W_p^p(\rho^y, \nu^y)]^{1/p} \tag{38}$$

$$= W_p^\mu(\eta, \rho) + W_p^\mu(\rho, \nu). \tag{39}$$

**Part (c).** We provide a counterexample. Fix any $u_0 \ne 0 \in U$ and $y_0, y_1 \in Y$ such that $y_0 \ne y_1$. Define $\mu = \frac{1}{2}\left(\delta_{y_0} + \delta_{y_1}\right)$. Set $u_k = (k+1)u_0$ for $k = 1, 2, \ldots$ and for each $k$, define two measures on $Y \times U$ by

$$\eta_k = \frac{1}{2}\left(\delta_{y_0 u_0} + \delta_{y_1 u_k}\right) \qquad \nu_k = \frac{1}{2}\left(\delta_{y_1 u_0} + \delta_{u_k y_0}\right). \tag{40}$$

It is clear that

$$W_p^{\mu, p}(\eta_k, \nu_k) = k^p |u_0|^p \qquad W_p^p(\eta_k, \nu_k) = \min\{k^p |u_0|^p, |y_1 - y_0|^p\}. \tag{41}$$

Moreover, as $k \to \infty$ we have $W_p^\mu(\mu_k, \nu_k) \to \infty$ but $W_p^p(\nu_k, \eta_k)$ remains bounded. See Figure 4.

**Part (d).** First, the unconditional distance $W_p(\eta, \nu)$ may be obtained via an unrestricted coupling in $\Pi(\eta, \nu)$, i.e. the set of all joint measures on $Y \times U$ having marginals $\eta, \nu$. Since $\Pi(\eta, \nu) \supseteq \Pi_Y(\eta, \nu)$, by part (a) we see that $W_p(\eta, \nu) \le W_p^\mu(\eta, \nu)$.

Let $\gamma^\star(y_0, u_0, y_1, u_1) = \gamma^{\star, y_1}(u_0, u_1)\delta(y_1 - y_0)\mu(y_1)$ be an optimal $\gamma^\star \in \Pi_Y(\eta, \nu)$. We claim that $\gamma(u_0, u_1) := \int_Y \gamma^{\star, y}(u_0, u_1) \, d\mu(y)$ couples $\pi_\#^U \eta$ and $\pi_\#^U \nu$. Let $\pi^0 : (u_0, u_1) \mapsto u_0$ be the projection onto the first coordinate of $U \times U$. Observe that for $\mu$-almost every $y$, we have that $\gamma^{\star, y} \in \Pi(\eta^y, \nu^y)$ is optimal, and, in particular, $\pi_\#^0 \gamma^{\star, y} = \eta^y$. Fix an arbitrary $\varphi \in C_b(U)$. We then have

$$\int_U \varphi(u_0) \, d\pi_\#^0 \gamma(u_0) = \int_{U^2} (\varphi \circ \pi^0) \, d\gamma(u_0, u_1) \tag{42}$$

$$= \int_Y \int_{U^2} (\varphi \circ \pi^0) \, d\gamma^{\star, y}(u_0, u_1) \, d\mu(y) = \int_Y \int_U \varphi(u_0) \, d\pi_\#^0 \gamma^{\star, y}(u_0) \, d\mu(y) \tag{43}$$

$$= \int_Y \int_U \varphi(u_0) \, d\eta^y(u_0) \, d\mu(y) = \int_{Y \times U} \varphi(u_0) \, d\eta(u_0, y) \tag{44}$$

$$= \int_{Y \times U} (\varphi \circ \pi^U) \, d\eta(u_0, y) = \int_U \varphi \, d\pi_\#^U \eta(u_0). \tag{45}$$

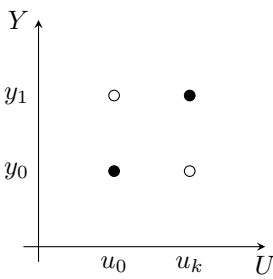

Figure 4: The counterexample in Proposition 2. The measure $\eta_k$ is shown in black and the measure $\nu_k$ is shown in white.

Thus $\pi^U_\# \gamma = \pi^U_\# \eta$. A similar argument shows that for the map $\pi^1 : (u_0, u_1) \mapsto u_1$ we have $\pi^1_\# \gamma = \pi^U_\# \nu$, so that $\gamma \in \Pi(\pi^U_\# \eta, \pi^U_\# \nu)$.

Now, as $\gamma^{\star, y_1}(u_0, u_1) \in \Pi(\eta^{y_1}, \nu^{y_1})$ is $\mu$-almost surely optimal in the usual Wasserstein sense,

$$W_p^{p,\mu}(\eta, \nu) = \int_Y \int_{U^2} |u_0 - u_1|^p \, d\gamma^{\star, y}(u_0, u_1) \, d\mu(y) \tag{46}$$

$$= \int_{U^2} |u_0 - u_1|^p \, d\gamma(u_0, u_1) \tag{47}$$

$$\geq W_p^p(\pi^U_\# \eta, \pi^U_\# \nu) \tag{48}$$

since $\gamma \in \Pi(\pi^U_\# \eta, \pi^U_\# \nu)$ is a coupling but potentially sub-optimal.

$\square$

## D.2 Geodesics

We now study the geodesics in the space $\mathbb{P}_p^\mu(Y \times U)$.

**Theorem 1** ($\mathbb{P}_p^\mu(Y \times U)$ is a Geodesic Space)
*For any $\eta, \nu \in \mathbb{P}_p^\mu(Y \times U)$, there exists a constant speed geodesic between $\eta$ and $\nu$.*

*Proof.* Write $\lambda_t : (Y \times U)^2 \to Y \times U$ for the linear interpolant

$$\lambda_t(y_0, u_0, y_1, u_1) = (ty_0 + (1-t)y_1, tu_0 + (1-t)u_1) \qquad 0 \leq t \leq 1. \tag{49}$$

Let $\gamma^\star \in \Pi_Y(\eta, \nu)$ be an optimal restricted coupling, and consider the path of measures in $\mathbb{P}_p(Y \times U)$ given by

$$\gamma_t = [\lambda_t]_\# \gamma^\star \qquad 0 \leq t \leq 1. \tag{50}$$

**Step one:** We check that for each $0 \leq t \leq 1$, we have $\gamma_t \in \mathbb{P}_p^\mu(Y \times U)$. That is, we need to check that for all Borel $A \subseteq Y$, we have $\gamma_t(A \times U) = \mu(A)$. Indeed, recall that restricted measures are concentrated on the set $\mathscr{C}$ (see Equation (19)). Thus,

$$\begin{aligned} \gamma_t(A \times U) &= \gamma^\star \left\{ \lambda_t^{-1}(A \times U) \right\} \\ &= \gamma^\star \left\{ (y, u_0, y, u_1) \mid y \in A \right\} \\ &= \pi^1_\# \gamma^\star(A) = (\pi^1 \circ \pi^{1,2})_\# \gamma^\star(A) \\ &= \pi^1_\# \eta(A) = \mu(A) \end{aligned}$$

i.e. $\gamma_t(A \times Y) = \mu(A)$ as claimed.

**Step two:** We show that $W_p^\mu(\gamma_t, \gamma_s) = |t - s| W_p^\mu(\eta, \nu)$. Set $\gamma_t^s := (\lambda_t, \lambda_s)_\# \gamma^\star$ for $0 \leq s < t \leq 1$. We claim $\gamma_t^s \in \Pi_Y(\gamma_t, \gamma_s)$. Indeed, we have $\pi^{1,2}_\# \gamma_t^s = \gamma_t$ because for all Borel $A \subseteq Y \times U$,

$$(\lambda_t, \lambda_s)_\# \gamma^\star (A \times Y \times U) = \gamma^\star \left( \lambda_t^{-1}(A) \right) = (\lambda_t)_\# \gamma^*(A). \tag{51}$$

An analogous calculation shows that $\pi_{\#}^{3,4}\gamma_t^s = \gamma_s$, so that $\gamma_t^s \in \Pi(\gamma_t, \gamma_s)$. We now check that $\gamma_t^s \in \mathcal{R}_Y(Y \times U)$. Indeed, suppose $E \subseteq Y \times U$ is a Borel set such that $E \cap \mathscr{C} = \varnothing$. In other words, for every $(y_0, u_0, y_1, u_1) \in E$ we have $y_0 \neq y_1$. Set $D := (\lambda_t, \lambda_s)^{-1}(E)$. We claim $D \cap \mathscr{C} = \varnothing$, so that

$$\gamma_t^s(E) = (\lambda_t, \lambda_s)_{\#}\gamma^{\star}(E) = \gamma^{\star}((\lambda_t, \lambda_s)^{-1}(E)) \tag{52}$$
$$= \gamma^{\star}(D \cap \mathscr{C}) = 0. \tag{53}$$

Indeed, if $c = (y, u_0, y, u_1) \in \mathscr{C}$, then

$$(\lambda_t, \lambda_s)(c) = (y, tu_0 + (1-t)u_1, y, su_0 + (1-s)u_1) \notin E \tag{54}$$
$$\implies c \notin (\pi_t, \pi_s)^{-1}(E). \tag{55}$$

Thus $\gamma_t^s \in \Pi_Y(\eta, \nu)$ as claimed. Now, we have

$$W_p^{\mu,p}(\gamma_t, \gamma_s) \leq \int_{(Y \times U)^2} d^p(y_0, u_0, y_1, u_1)\, d\lambda_t^s(y_0, u_0, y_1, u_1)$$

$$= \int_{(Y \times U)^2} d^p(\lambda_t(y_0, u_0, y_1, u_1), \lambda_s(y_0, u_0, y_1, u_1))\, d\gamma^{\star}(y_0, u_0, y_1, u_1)$$

$$= \int_{(Y \times U)^2} \left(|(t-s)(y_0 - y_1)|^2 + |(t-s)(u_0 - u_1)|^2\right)^{p/2}\, d\gamma^{\star}(y_0, u_0, y_1, u_1)$$

$$= |t-s|^p \int_{(Y \times U)^2} d^p(y_0, u_0, y_1, u_1)\, d\gamma^{\star}(y_0, u_0, y_1, u_1)$$

$$= |t-s|^p W_p^{\mu,p}(\eta, \nu).$$

Conversely, an application of the previous inequality and the triangle inequality show that for $0 \leq s \leq t \leq 1$,

$$W_p^{\mu}(\eta, \nu) \leq W_p^{\mu}(\eta, \gamma_s) + W_p^{\mu}(\gamma_s, \gamma_t) + W_p^{\mu}(\gamma_t, \nu) \tag{56}$$
$$\leq s W_p^{\mu}(\eta, \nu) + W_p^{\mu}(\gamma_s, \gamma_t) + (1-t)W_p^{\mu}(\eta, \nu). \tag{57}$$

Rearranging the previous inequality implies $|t-s|W_p^{\mu}(\eta, \nu) \leq W_p^{\mu}(\gamma_s, \gamma_t)$ for all $s, t \in [0, 1]$, and hence $W_p^{\mu}(\gamma_t, \gamma_s) = |t-s|^p W_p^{\mu}(\eta, \nu)$. $\qquad\square$

**Theorem 2** (Conditional McCann Interpolants)
*Fix $\eta, \nu \in \mathbb{P}_p^{\mu}(Y \times U)$. Suppose $T^{\star}(y, u) = (y, T_{\mathcal{U}}^{\star}(y, u))$ is an injective triangular map solving the conditional Monge problem* (1). *Define the maps $T_t : Y \times U \to Y \times U$ for $0 \leq t \leq 1$ via $T_t = (1-t)I + tT^{\star}$, and define the curve of measures $\gamma_t = [T_t]_{\#}\eta \in \mathbb{P}_p^{\gamma}(Y \times U)$. Then,*

(a) *$(\gamma_t)$ is absolutely continuous and a constant speed geodesic between $\eta, \nu$*

(b) *The vector field $v_t(T_t^{\star}(y, u)) = (0, T_{\mathcal{U}}^{\star}(y, u) - u)$ generates the path $\gamma_t$, in the sense that $(\gamma_t, v_t)$ solve the continuity equation* (9).

*Proof.* Consider the function $w_t : Y \times U \to U$ given by

$$w_t(y, u) = (0, T_U^{\star}(y, u) - u) = (0, w_{t,U}(y, u)) \tag{58}$$

and note this is precisely $w_t(y, u) = \partial_t T_t^{\star}(y, u)$. Define the vector field

$$v_t(y, u) = \left(w_t \circ T_t^{\star, -1}\right)(y, u) = \left(0, (w_{t,\mathcal{U}} \circ T_{t,\mathcal{U}}^{\star, -1})(y, u)\right). \tag{59}$$

For any $\varphi \in \mathrm{Cyl}(Y \times U)$, we have

$$\frac{d}{dt}\int_{Y \times U} \varphi(y, u)\, d\gamma_t(y, u) = \frac{d}{dt}\int_{Y \times U} \varphi(y, u)\, d[T_t]_{\#}\eta(y, u) \tag{60}$$

$$= \frac{d}{dt}\int_{Y \times U} \varphi(y, T_{t,U}^{\star}(y, u))\, d\eta(y, u) \tag{61}$$

$$= \int_{Y \times U} \langle \nabla\varphi(y, T_{t,U}^{\star}(y, u)), w_t(y, u)\rangle\, d\eta(y, u) \tag{62}$$

$$= \int_{Y \times U} \langle \nabla\varphi(y, u), v_t(y, u)\rangle\, d\gamma_t(y, u) \tag{63}$$

which shows that $(\gamma_t, v_t)$ solve the continuity equation.

Now, note that for $0 \le a \le b \le 1$, we have

$$\int_a^b \|v_t\|_{L^p(\gamma_t, Y \times U)} \, \mathrm{d}t = \int_a^b \left( \int_{Y \times U} \left| w_t \circ T_t^{\star, -1} \right|^p (y, u) \, \mathrm{d}\gamma_t(y, u) \right)^{1/p} \mathrm{d}t \qquad (64)$$

$$= \int_a^b \left( \int_{Y \times U} |w_t|^p (y, u) \, \mathrm{d}\eta(y, u) \right)^{1/p} \mathrm{d}t \qquad (65)$$

$$= \int_a^b \left( \int_{Y \times U} |u - T_U^\star(y, u)|^p (y, u) \, \mathrm{d}\eta(y, u) \right)^{1/p} \mathrm{d}t \qquad (66)$$

$$= (b - a) W_p^\mu(\eta, \nu). \qquad (67)$$

In particular, $\int_0^1 \|v_t\|_{L^p(\gamma_t, Y \times U)} \, \mathrm{d}t < \infty$ and so by Theorem 4 $(\gamma_t)$ is absolutely continuous. A similar calculation shows that $(b - a) W_p^\mu(\eta, \nu) = W_p^\mu(\gamma_b, \gamma_a) = \int_a^b |\gamma'(t)|$, where the last line follows from the absolute continuity of $\gamma_t$. Thus, $\|v_t\|_{L^p(\gamma_t, Y \times U)} = |\gamma'|(t)$ for almost every $t \in [0, 1]$ by Lebesgue differentiation.

$\qquad \square$

# E    Proofs: Section 5

In this section, we provide proofs of all claims made in Section 5.

## E.1    Continuity Equation

We begin with a lemma that is used in the proof of Theorem 4. Informally, a solution to the continuity equation with a triangular vector field will result in the conditional measures almost surely satisfying the continuity equation as well.

**Lemma 1** (Triangular Vector Fields Preserve Conditionals)
*Suppose $v_t(y, u) = (0, v_t^U(y, u))$ is triangular and that $(\gamma_t) \subset \mathbb{P}_p^\mu(Y \times U)$ is a path of measures such that $(v_t, \gamma_t)$ satisfy the continuity equation in the sense of distributions. Then, it follows that for $\mu$-almost every $y \in Y$, we have $\partial_t \gamma_t^y + \nabla \cdot (v_t^U(y, -)\gamma_t^y) = 0$.*

*Proof.* Fix any $\varphi \in \mathrm{Cyl}(U \times I)$. Suppose $\psi \in \mathrm{Cyl}(Y)$ is given, and note that $\psi(y)\varphi(u, t) \in \mathrm{Cyl}(Y \times U \times I)$. As $(v_t, \gamma_t)$ solve the continuity equation, it follows from the triangular structure of $v_t$ that upon testing against $\psi\varphi$ we have

$$\int_I \int_Y \psi(y) \int_U \left( \partial_t \varphi(u, t) + \langle v_t^U(y, u), \nabla_u \varphi(u, t) \rangle \right) \mathrm{d}\gamma_t^y(u) \, \mathrm{d}\mu(y) \, \mathrm{d}t = 0. \qquad (68)$$

Because $\psi(y) \in \mathrm{Cyl}(Y)$, it is of the form $\rho(\pi(y))$ where $\pi : Y \to \mathbb{R}^k$ for some $k \ge 1$ and $\rho \in C_c^\infty(\mathbb{R}^k)$. Taking $\rho$ to be a sequence of smooth approximations to the indicator function of an arbitrary rectangle $E = E_1 \times E_2 \times \cdots \times E_k \subseteq \mathbb{R}^k$, we see

$$\int_{\pi^{-1}(E)} \int_I \int_U \left( \partial_t \varphi(u, t) + \langle v_t^U(y, u), \nabla_u \varphi(u, t) \rangle \right) \mathrm{d}\gamma_t^y(u) \, \mathrm{d}t \, \mathrm{d}\mu(y) = 0. \qquad (69)$$

As $Y$ is separable, the Borel $\sigma$-algebra on $Y$ is generated by the cylinder sets, i.e. those which are precisely of the form $\pi^{-1}(E)$ for some finite-dimensional rectangle $E$. We have thus shown that for an arbitrary Borel measurable set $E \subseteq Y$,

$$\int_E \int_I \int_U \left( \partial_t \varphi(u, t) + \langle v_t^U(y, u), \nabla_u \varphi(u, t) \rangle \right) \mathrm{d}\gamma_t^y(u) \, \mathrm{d}t \, \mathrm{d}\mu(y) = 0. \qquad (70)$$

From this, it follows that

$$\int_I \int_U \left( \partial_t \varphi(u, t) + \langle v_t^U(y, u), \nabla_u \varphi(u, t) \rangle \right) \mathrm{d}\gamma_t^y(u) \, \mathrm{d}t = 0 \qquad \mu\text{-almost every } y. \qquad (71)$$

$\qquad \square$

## E.2 Absolutely Continuous Curves

We now proceed to prove the main results of this section. First, we introduce some preliminary notions. We define the map $j_q : L^q(\gamma, Y \times U) \to L^p(\gamma, Y \times U)$ for $1/p + 1/q = 1$ via

$$j_q(w) = \begin{cases} |w|^{q-2}w & w \neq 0 \\ 0 & w = 0 \end{cases} \tag{72}$$

which is the Fréchet differential of the convex functional $\frac{1}{q} \|w\|_{L^q(\gamma, Y \times U)}^q$. A straightforward calculation shows that this map satisfies

$$\|j_q(w)\|_{L^p(\gamma, Y \times U)}^p = \|w\|_{L^q(\gamma, Y \times U)}^q = \int_{Y \times U} \langle j_q(w), w \rangle \, d\gamma(y, u). \tag{73}$$

See also Ambrosio et al. [2005, Chapter 8].

**Theorem 3** (Absolutely Continuous Curves in $\mathbb{P}_p^\mu(Y \times U)$)
*Let $I \subset \mathbb{R}$ be an open interval, and suppose $\gamma_t : I \to \mathbb{P}_p^\mu(Y \times U)$ is an absolutely continuous in the $W_p^\mu$ metric with $|\gamma'|(t) \in L^1(I)$. Then, there exists a Borel vector field $v_t(y, u)$ such that*

- *(a) $v_t$ is triangular*

- *(b) $v_t \in L^p(\gamma_t, Y \times U)$ and $\|v_t\|_{L^p(\gamma_t, Y \times U)} \leq |\gamma'|(t)$ for a.e. $t$*

- *(c) $(v_t, \gamma_t)$ solve the continuity equation in the sense of distributions.*

*Proof.* Assume without loss of generality that $|\gamma'|(t) \in L^\infty(I)$ and that $I = (0, 1)$ [Ambrosio et al., 2005, Lemma 1.1.4, Lemma 8.1.3]. Fix any $\varphi \in \mathrm{Cyl}(Y \times U)$. For $s, t \in I$ there exists an optimal triangular coupling $\gamma_{st} \in \Pi_Y(\gamma_s, \gamma_t)$. By Hölder's inequality,

$$|\gamma_t(\varphi) - \gamma_s(\varphi)| \leq \mathrm{Lip}(\varphi) W_p^\mu(\gamma_s, \gamma_t). \tag{74}$$

It follows that $t \mapsto \gamma_t(\varphi)$ is absolutely continuous. We can introduce the upper semicontinuous and bounded map

$$H(y_0, u_0, y_1, u_1) = \begin{cases} |\nabla \varphi(y_0, u_0)| & (y_0, u_0) = (y_1, u_1) \\ \frac{|\varphi(y_0, u_0) - \varphi(y_1, u_1)|}{|(y_0, u_0) - (y_1, u_1)|} & (y_0, u_0) \neq (y_1, u_1) \end{cases}. \tag{75}$$

For $|h|$ sufficiently small, choose any optimal coupling $\gamma_{(s+h)h} \in \Pi_Y(\gamma_{s+h}, \gamma_s)$ and note that

$$\frac{|\gamma_{s+h}(\varphi) - \gamma_s(\varphi)|}{|h|} \leq \frac{1}{|h|} \int_{(Y \times U)^2} |(y_0, u_0) - (y_1, u_1)| H(y_0, u_0, y_1, u_1) \, d\gamma_{(s+h)s} \tag{76}$$

$$\leq \frac{W_p^\mu(\gamma_{s+h}, \gamma_s)}{|h|} \left( \int_{(Y \times U)^2} H^q(y_0, u_0, y_1, u_1) \, d\gamma_{(s+h)h,s} \right)^{1/q}. \tag{77}$$

If $t$ is a point of metric differentiability for $t \mapsto \gamma_t$, note that $\gamma_{(t+h)t} \to (I, I)_\# \gamma_t$ narrowly, where $I$ is the identity map on $Y \times U$. Moreover, since $\gamma_t \in \mathbb{P}_p^\mu(Y \times U)$, it follows that on the diagonal we have that almost surely $H(y_0, u_0, y_0, u_1) = \iota(|\nabla_u \varphi(y_0, u_0)|)$. Thus,

$$\limsup_{h \to 0} \frac{|\gamma_{t+h}(\varphi) - \gamma_t(\varphi)|}{|h|} \leq |\gamma'|(t) \left( \int_{Y \times U} |H|^q(y_0, u_0, y_0, u_0) \, d\gamma_t(y_0, u_0) \right)^{1/q} \tag{78}$$

$$= |\gamma'|(t) \|\iota(\nabla_u \varphi)\|_{L^q(\gamma_t, Y \times U)} = |\gamma'|(t) \|\nabla_u \varphi\|_{L^q(\gamma_t, U)}. \tag{79}$$

Taking $Q = Y \times U \times I$ and $\gamma = \int \gamma_t \, dt$, fix any $\varphi \in \mathrm{Cyl}(Q)$. We have that

$$\int_Q \partial_s \varphi(y, u, s) \, d\gamma(y, u, s)$$
$$= \lim_{h \downarrow 0} \int_I \frac{1}{h} \left( \int_{Y \times U} \varphi(y, u, s) \, d\gamma_s(y, u) - \int_{(Y \times U)} \varphi(y, u, s) \, d\gamma_{s+h}(y, u) \right) ds. \tag{80}$$

An application of Fatou's Lemma, Equation (78), and Hölder's inequality gives us

$$\left| \int_Q \partial_s \varphi(y,u,s) \, \mathrm{d}\gamma(y,u,s) \right| \leq \left( \int_J |\gamma'|(s) \, \mathrm{d}s \right)^{1/p} \left( \int_Q |\nabla_u \varphi(y,u,s)|^q \, \mathrm{d}\mu(y,u,s) \right)^{1/q} \quad (81)$$

for any interval $J \subset I$ with $\operatorname{supp} \varphi \subset J \times Y \times U$.

Fix the subspace

$$V = \{ \iota(\nabla_u \varphi(y,u,s)) : \varphi \in \operatorname{Cyl}(Q) \} \subseteq Y \times U \quad (82)$$

and denote by $\overline{V}$ its $L^q(\gamma, Y \times U \times I)$ closure. Define the linear functional $L : V \to \mathbb{R}$ via

$$L(\nabla_u \varphi) = - \int_Q \partial_s \varphi(y,u,s) \, \mathrm{d}\gamma(y,u,s) \quad (83)$$

and note that Equation (81) implies that $L$ is a bounded linear functional on $V$. Thus (by Hahn-Banach and the fact that $V \subseteq \overline{V}$ is dense) we may uniquely extend $L$ to $\overline{V}$. We thus have a convex minimization problem

$$\min_{w \in \overline{V}} \frac{1}{q} \int_Q |w(y,u,s)|^q \, \mathrm{d}\gamma(y,u,s) - L(w) \quad (84)$$

which admits the unique solution $w$ such that $j_q(w) - L = 0$. In particular, the estimate (81) shows that the above functional is coercive and hence admits a minimizer which we may obtain via its differential as a consequence of convexity. Thus, we obtain a triangular vector field $v = j_q(w)$ such that for all $\varphi \in \operatorname{Cyl}(Q)$,

$$\langle v, \nabla \varphi \rangle = \int_Q \langle v(y,u,s), \nabla \varphi(y,u,s) \rangle \, \mathrm{d}\gamma(y,u,s) = \langle L, \nabla \varphi \rangle = - \int_Q \partial_s \varphi(y,u,s) \, \mathrm{d}\gamma(y,u,s). \quad (85)$$

This precisely shows that $(v_t, \gamma_t)$ is a triangular distributional solution to the continuity equation.

Now, choose any interval $J \subset I$ and choose a sequence $\eta^k \in C_c^\infty(J)$, with $0 \leq \eta^k \leq 1$ and $\eta_k \to \mathbb{1}_J$ as $k \to \infty$. Moreover choose a sequence $(\nabla_u \varphi_n) \subset V$ converging to $w = j_p(v)$ in $L^q(\gamma, Q)$. Our previous calculations give

$$\int_Q \eta^k(s) |v(y,u,s)|^p \, \mathrm{d}\gamma(y,u,s) = \int_Q \eta^k(s) \langle v, w \rangle \, \mathrm{d}\gamma = \lim_{n \to \infty} \int_Q \eta^k \langle v, \nabla_u \varphi_n \rangle \, \mathrm{d}\gamma \quad (86)$$

$$= \lim_{n \to \infty} \langle L, \nabla_u(\eta^k \varphi_n) \rangle \leq \left( \int_J |\gamma'|^p(s) \, \mathrm{d}s \right)^{1/p} \left( \int_{J \times Y \times U} |v|^p \, \mathrm{d}\gamma \right)^{1/p}. \quad (87)$$

Taking $k \to \infty$ we see that

$$\int_J \int_{Y \times U} |v_t(y,u)|^p \, \mathrm{d}\gamma_t(y,u) \, \mathrm{d}t \leq \int_J |\gamma'|^p(s) \, \mathrm{d}s \quad (88)$$

and since $J \subset I$ was arbitrary, we conclude

$$\| v_t \|_{L^p(\gamma_t, Y \times U)} \leq |\gamma'|(t) \qquad \text{a.e.-} t. \quad (89)$$

$\square$

We now prove, in some sense, a converse of the previous theorem.

**Theorem 4** (Continuous Curves Generated by Triangular Vector Fields)
*Suppose that $\gamma_t : I \to \mathbb{P}_p^\mu(Y \times U)$ is narrowly continuous and $(v_t)$ is a triangular vector field such that $(\gamma_t, v_t)$ solve the continuity equation with $\| v_t \|_{L^p(\gamma_t, Y \times U)} \in L^1(I)$. Then, $\gamma_t : I \to \mathbb{P}_p^\mu(Y \times U)$ is absolutely continuous in the $W_p^\mu$ metric and $|\gamma'|(t) \leq \| v_t \|_{L^p(\mu, Y \times U)}$ for almost every $t$.*

*Proof.* We first assume that $U$ is finite dimensional. Our strategy is to check the hypotheses necessary for Ambrosio et al. [2005, Theorem 8.3.1] to hold for $\mu$-almost every $y$, followed by an application of this theorem. By Lemma 1, for $\mu$-almost every $y$ we have that $(\gamma_t^y, v_t^U(y, -))$ solve the continuity equation distributionally on $I \times U$.

By Jensen's inequality (and the assumption $p \geq 1$) we see

$$\int_I \|v_t\|_{L^p(\gamma_t, Y \times U)} \, \mathrm{d}t = \int_I \mathbb{E}_{y \sim \mu} \left[ \|v_t^U(y, -)\|_{L^p(\gamma_t^y, U)}^p \right]^{1/p} \mathrm{d}t \tag{90}$$

$$\geq \int_I \mathbb{E}_{y \sim \mu} \left[ \|v_t^U(y, -)\|_{L^p(\gamma_t^y, U)} \right] \mathrm{d}t \tag{91}$$

$$= \mathbb{E}_{y \sim \mu} \left[ \int_I \|v_t^U(y, -)\|_{L^p(\gamma_t^y, U)} \, \mathrm{d}t \right]. \tag{92}$$

Since the first term is finite, it follows that

$$\|v_t^U(y, -)\|_{L^p(\gamma_t^y, U)} \in L^1(I) \qquad \mu\text{-almost every } y. \tag{93}$$

Now Ambrosio et al. [2005, Lemma 8.1.2] shows that for $\mu$-almost every $y$ we have that $(\gamma_t^y)$ admits a narrowly continuous representative $(\tilde{\gamma}_t^y)$ with $\tilde{\gamma}_t^y = \gamma_t^y$ for almost every $t$. It follows from Ambrosio et al. [2005, Theorem 8.3.1] that for any $t_1 \leq t_2$ in $I$, we have

$$W_p^p(\tilde{\gamma}_{t_1}^y, \tilde{\gamma}_{t_2}^y) \leq (t_2 - t_1)^{p-1} \int_{t_1}^{t_2} |v_t^U(y, u)|^p \, \mathrm{d}\tilde{\gamma}_t^y(u) \, \mathrm{d}t \tag{94}$$

$$= (t_2 - t_1)^{p-1} \int_{t_1}^{t_2} |v_t^U(y, u)|^p \, \mathrm{d}\gamma_t^y(u) \, \mathrm{d}t \tag{95}$$

where the second line follows as $\tilde{\gamma}_t^y = \gamma_t^y$ for almost every $t$.

Let $\tilde{\gamma}_t = \int_Y \tilde{\gamma}_t^y \, \mathrm{d}\mu(y)$ be the measure obtained via marginalizing over the $Y$-variables. Taking an expectation over $y \sim \mu$, the previous inequality shows us that

$$\frac{W_p^{\mu, p}(\tilde{\gamma}_{t_1}, \tilde{\gamma}_{t_2})}{(t_2 - t_1)^p} \leq \frac{1}{t_2 - t_1} \int_{t_1}^{t_2} \|v_t\|_{L^p(\gamma_t, Y \times U)}^p \, \mathrm{d}t. \tag{96}$$

Now, note that $t_1$ is almost surely a Lebesgue point of the right-hand side and $\tilde{\gamma}_{t_1} = \gamma_{t_1}$. Taking $t_2 \to t_1$ along a sequence where $\tilde{\gamma}_{t_2} = \gamma_{t_2}$ shows us that

$$|\gamma'|(t) \leq \|v_t\|_{L^p(\gamma_t), Y \times U} \tag{97}$$

for almost every $t \in I$.

In the case that $U$ is infinite dimensional, fix any $y \in Y$ such that Lemma 1 holds (which is of full measure) and fix a countable orthonormal basis $(e_k)$ for $U$. Set $\pi^d : U \to \mathbb{R}^d$ to be the projection operator for this basis, i.e. $u \mapsto (\langle u, e_1 \rangle, \ldots, \langle u, e_d \rangle)$. We consider the collection of finite dimensional conditional measures $\gamma_t^{d, y} = \pi_\#^d \gamma_t^y$. By the same argument in Ambrosio et al. [2005, Theorem 8.3.1], there exists a vector field $v_t^{d, y}$ on $\mathbb{R}^d$ such that $(\gamma_t^{d, y}, v_t^{d, y})$ solve the continuity equation and

$$\|v_t^{d, y}\|_{L^p(\gamma_t^{d, y}, \mathbb{R}^d)} \leq \|v_t^U(y, -)\|_{L^p(\gamma_t^y, U)}. \tag{98}$$

It follows from the finite-dimensional case above that for almost every $t_1 \leq t_2$, we have

$$W_p^p(\gamma_{t_1}^{d, y}, \gamma_{t_2}^{d, y}) \leq (t_2 - t_1)^{p-1} \int_{t_1}^{t_2} \|v_t^U(y, -)\|_{L^p(\gamma_t^y, U)}^p \, \mathrm{d}t. \tag{99}$$

Let $\hat{\gamma}_t^{y, d} = (\pi^d)_\#^\star \gamma_t^{y, d}$ where $(\pi^d)^\star : \mathbb{R}^d \to U$ maps $z \mapsto \sum_{k=1}^d z_k e_k$. As $d \to \infty$ we have $\hat{\gamma}_t^{d, y} \to \gamma_t^y$ narrowly for all $t \in I$. Since $(\pi^d)^\star$ is an isometry, Ambrosio et al. [2005, Lemma 7.1.4] shows that

$$W_p^p(\gamma_{t_1}^y, \gamma_{t_2}^y) \leq \liminf_{d \to \infty} W_p^p(\gamma_{t_1}^{d, y}, \gamma_{t_2}^{d, y}) \leq (t_2 - t_1)^{p-1} \int_{t_1}^{t_2} \|v_t^U(y, -)\|_{L^p(\gamma_t^y, U)}^p \, \mathrm{d}t. \tag{100}$$

Now, integration with respect to $\mathrm{d}\mu(y)$ yields

$$W_p^{p,\mu}(\gamma_{t_1}, \gamma_{t_2}) \leq (t_2 - t_1)^{p-1} \int_{t_1}^{t_2} \|v_t\|_{L^p(\gamma_t, Y \times U)}^p \, \mathrm{d}t. \tag{101}$$

Taking $t_2 \to t_1$ shows that for almost every $t$ we have

$$|\gamma'|(t) \leq \|v_t\|_{L^p(\gamma_t, Y \times U)}. \tag{102}$$

$\square$

Together, Theorem 3 and Theorem 4 give us a dynamical interpretation of the conditional Wasserstein distance. The following result is a conditional analogue of the well-known Benamou-Brenier Theorem [Benamou and Brenier, 2000]. Here, we note that the following proof follows the standard proof closely – the main legwork in obtaining this conditional generalization is through the previous two theorems.

**Theorem 5** (Conditional Benamou-Brenier)
*Let* $1 < p < \infty$. *For any* $\eta, \nu \in \mathbb{P}_p^\mu(Y \times U)$, *we have*

$$W_p^{p,\mu}(\eta, \nu) = \min_{(\gamma_t, v_t)} \left\{ \int_0^1 \|v_t\|_{L^p(\mu_t)}^p \, \mathrm{d}t \mid (v_t, \gamma_t) \text{ solve (9)}, \gamma_0 = \eta, \gamma_1 = \nu, \text{ and } v_t \text{ is triangular} \right\}.$$

*Proof.* Write $M$ for the infimum on the right-hand side.

First, suppose that $(v_t, \mu_t)$ are admissible and $\int_0^1 \|v_t\|_{L^p(\mu_t)} < \infty$. It follows from Theorem 4 that $(\gamma_t)$ is an absolutely continuous curve in $\mathbb{P}_p^\mu(Y \times U)$ and $\|v_t\|_{L^p(\mu_t, Y \times U)} \geq |\gamma'|(t)$. Thus,

$$W_p^{\mu,p}(\eta, \nu) \leq \left( \int_0^1 |\gamma'|(t) \, \mathrm{d}t \right)^p \leq \int_0^1 \|v_t\|_{L^p(\mu_t, Y \times U)}^p \, \mathrm{d}t \leq M. \tag{103}$$

Conversely, by Theorem 1 there exists a constant speed geodesic $(\gamma_t) \subset \mathbb{P}_p^\mu(Y \times U)$ connecting $\eta$ and $\nu$. Recall that constant speed geodesics are absolutely continuous. By Theorem 3, there exists a Borel triangular vector field $v_t$ such that $(v_t, \gamma_t)$ solve the continuity equation, and moreover $\|v_t\|_{L^p(\mu_t, Y \times U)} \leq |\gamma'|(t)$. In fact, because $(v_t, \gamma_t)$ solve the continuity equation, Theorem 4 yields that $\|v_t\|_{L^p(\mu_t, Y \times U)} = |\gamma'|(t)$.

Since $\gamma_t$ is a constant speed geodesic in $\mathbb{P}_p^\mu(Y \times U)$, it follows that $|\mu'|(t) = W_p^\mu(\eta, \nu)$ for almost every $t \in (0, 1)$. Hence,

$$W_p^{\mu,p}(\eta, \nu) = \int_0^1 |\gamma'|(t)^p \, \mathrm{d}t = \int_0^1 \|v_t\|_{L^p(\gamma_t, Y \times U)}^p \geq M. \tag{104}$$

Thus, $W_p^{\mu,p}(\eta, \nu) = M$ as desired. $\square$

# F  Experiment Details

In this section, we provide additional details regarding all of our experiments, as well as additional results not contained within the main paper. All models can be trained on a single GPU with less than 24 GB of memory, and our experiments were parallelized over 8 such GPUs on a local server. We first describe our setting for the 2D and Lotka-Volterra experiments, as these share a similar setup. Details for the Darcy flow inverse problem are described in the corresponding section.

**Models.**  For FM and COT-FM, our model architecture is an MLP with SeLU activations [Klambauer et al., 2017]. Time conditioning is achieved by concatenating the time variable as an input to the network. The covariance operator $C$ chosen in the path of measures in Equation (10) is taken to be $C = \sigma^2 I$ where $\sigma$ is a hyperparameter.

Our implementation of FM is adapted from the `torchcfm` package Tong et al. [2023], available under the MIT License. For PCP-Map and COT-Flow, we adapt the open-source implementations from Wang et al. [2023], available under the MIT License.

Table 4: Hyperparameter grid used for random search of the FM and COT-FM models on the 2D and Lotka-Volterra datasets.

| Hyperparameter | Description | Values |
|---|---|---|
| $\epsilon$ | COT coupling strength | [1e-6, 1e-4, 1e-2, 1e-1] |
| $\sigma$ | Variance for $C = \sigma^2 I$ in (10) | [1e-3, 1e-2, 1e-1, 5e-1] |
| Batch Size | Training batch size | [256, 512, 1024] |
| Width | Layer width in MLP | [256, 512, 1024, 2048] |
| LR | Learning rate | [1e-4, 3e-4, 7e-4, 1e-3] |
| Layers | Number of MLP layers | [4, 6, 8] |

**Training and Model Selection.** Hyperparameter tuning of the PCP-Map and COT-Flow models was performed directly using the code of Wang et al. [2023], essentially implementing grid-search with an early stopping procedure. We refer to the paper and codebase of Wang et al. [2023] for further details. For COT-FM and FM, we perform a random grid search over 100 hyperparameter settings using the grid described in Table 4. For all model types, we select the best model used to generate the results in the paper as the training checkpoint that resulted in the lowest $W_2$ error to the joint target distribution on a held-out validation set. For training, we use the Adam optimizer where we only tune the learning rate, leaving all other settings as their defaults in `pytorch`.

## F.1    2D Synthetic Data

**Data Generation.** This experiment consists of four 2D synthetic datasets, where $Y = U = \mathbb{R}$. The datasets moons, circles, swissroll are available through `scikit-learn` [Pedregosa et al., 2011]. The moons dataset is generated with `noise=0.05` followed by standard scaling with a mean of $m = (0.5, 0.25)$ and standard deviation of $\sigma = (0.75, 0.25)$. The circles dataset is generated with `factor=0.5` and `noise=0.05`. The swissroll dataset is generated with `noise=0.75`, followed by projection to the first two coordinates and re-scaling by a factor of 12. All other unstated parameters are left as their default values. We use the code available from Hosseini et al. [2023] to generate the checkerboard dataset. For all datasets, we generate a training set (i.e., samples from the target distribution) of $20,000$ samples and $1,000$ held-out validation samples for model selection. Means and standard deviations in Table 1 are reported across five independent testing sets of 5,000 samples for the best representative of each model type.

In COT-FM, to generate samples from the source distribution, we sample an additional $20,000$ points from the target distribution and keep only the $Y$ coordinates. This ensures that the source and target have equal $Y$ marginals. During training, standard Gaussian noise $\mathcal{N}(0,1)$ is sampled for the $U$ coordinate of these source points at each minibatch.

We use minibatch COT couplings [Tong et al., 2023] in this experiment as computing the full COT plan was prohibitively expensive in terms of memory usage. However, we note that we use large batch sizes, meaning that the COT plan we find in this way should not be too far from optimal. All couplings are computed using the `POT` Python package [Flamary et al., 2021].

## F.2    Lotka-Volterra Dynamical System

**Data Generation.** We adopt the settings of Alfonso et al. [2023] for this experiment. As described in the main paper, we assume $p(0) = (30, 1)$ and that $\log(u) \sim \mathcal{N}(m, 0.5I)$ with $m = (-0.125, -3, -0.125, -3)$. Given parameters $u \in \mathbb{R}^4_{\geq 0}$, we simulate Equation (13) for $t \in \{0, 2, \ldots, 20\}$ to obtain a solution $z(u) \in \mathbb{R}^{22}_{\geq 0}$. An observation $y \in \mathbb{R}^{22}_{\geq 0}$ is obtained by the addition of log-normal noise, i.e. $\log(y) \sim \mathcal{N}(\log(z(u)), 0.1I)$. We thus may simulate many $(y, u)$ pairs from the target measure for training.

We generate a training set of $10,000$ $(y, u)$ pairs using the procedure described above and a held-out validation set of $10,000$ $(y, u)$ pairs for model selection. Means and standard deviations in Table 2 are reported across five independent testing sets of 5,000 samples for the best representative of each model type. Figure 2 and Figures 9, 8, 7, 10 show $10,000$ samples from each model, as well as $10,000$ samples from the differential evolution Metropolis MCMC sampler [Braak, 2006] after a

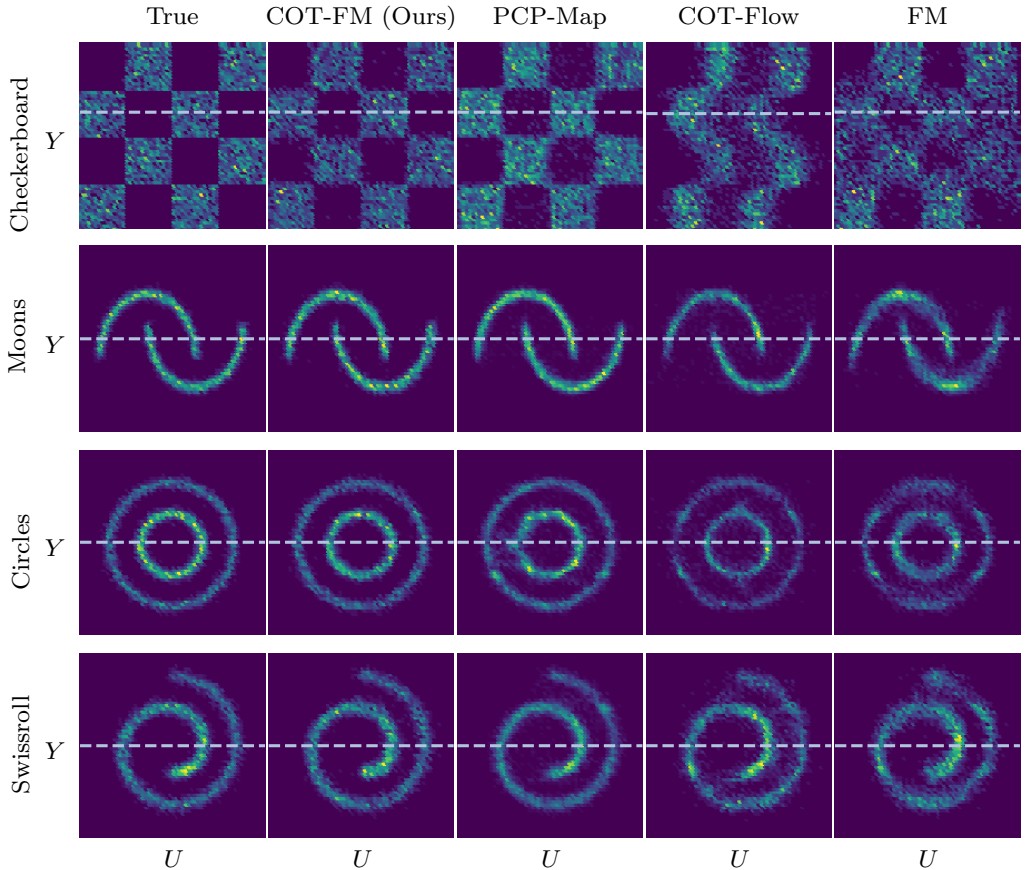

Figure 5: Samples from the ground-truth joint target distribution and the various models for the 2D datasets. Samples from COT-FM more closely match the ground-truth distribution than the baselines. A common failure mode for the baselines is to generate samples from regions with zero support under the true data distributions. Table 1 contains a quantitative evaluation.

burn-in of $50,000$ samples. This is implemented through the `PyMC` Python package [Abril-Pla et al., 2023].

For COT-FM we use the full COT couplings, i.e. without minibatches. This is available to use due to the smaller size of the training set used in this experiment. The COT couplings are computed in the same way as the previous section, and as described in Section 7.

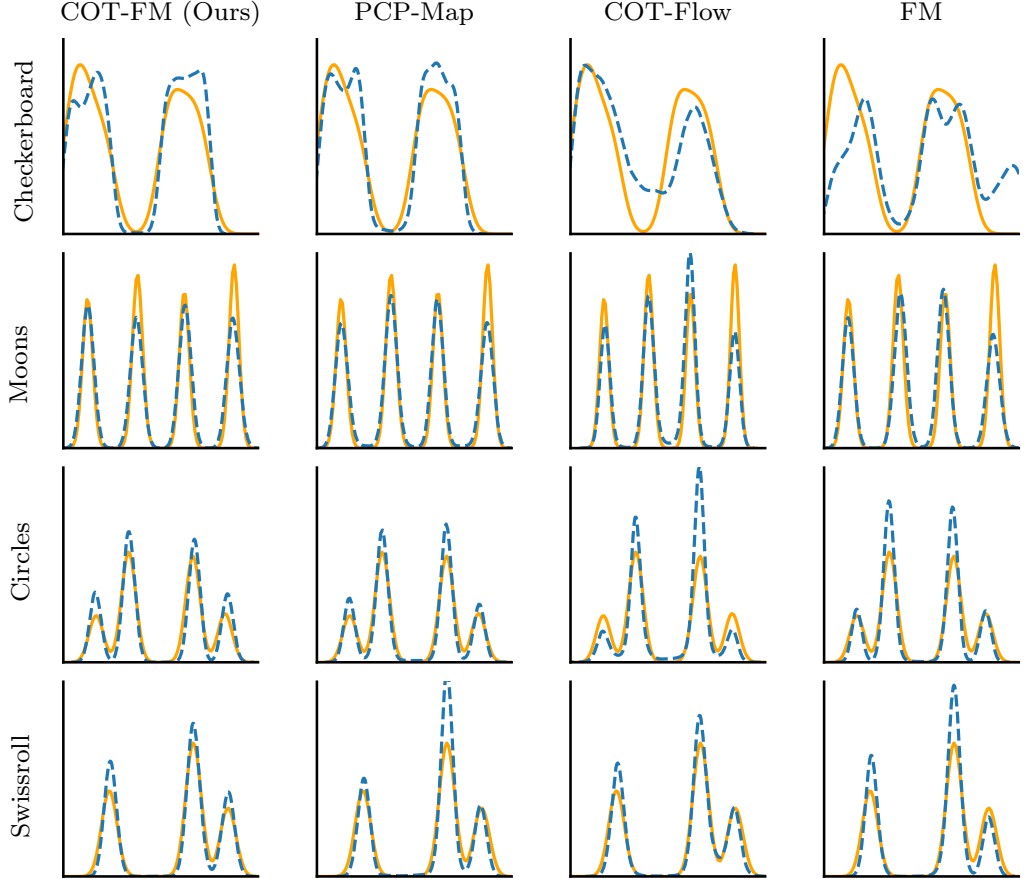

Figure 6: Conditional KDEs shown for each of the methods on the 2D datasets. The conditioning variable $y$ is fixed at the horizontal dashed line shown in Figure 5. In all plots, the orange solid line indicates the CKDE of the ground-truth joint samples. In each column, the dashed blue line indicates the CKDE of samples generated from the respective method.

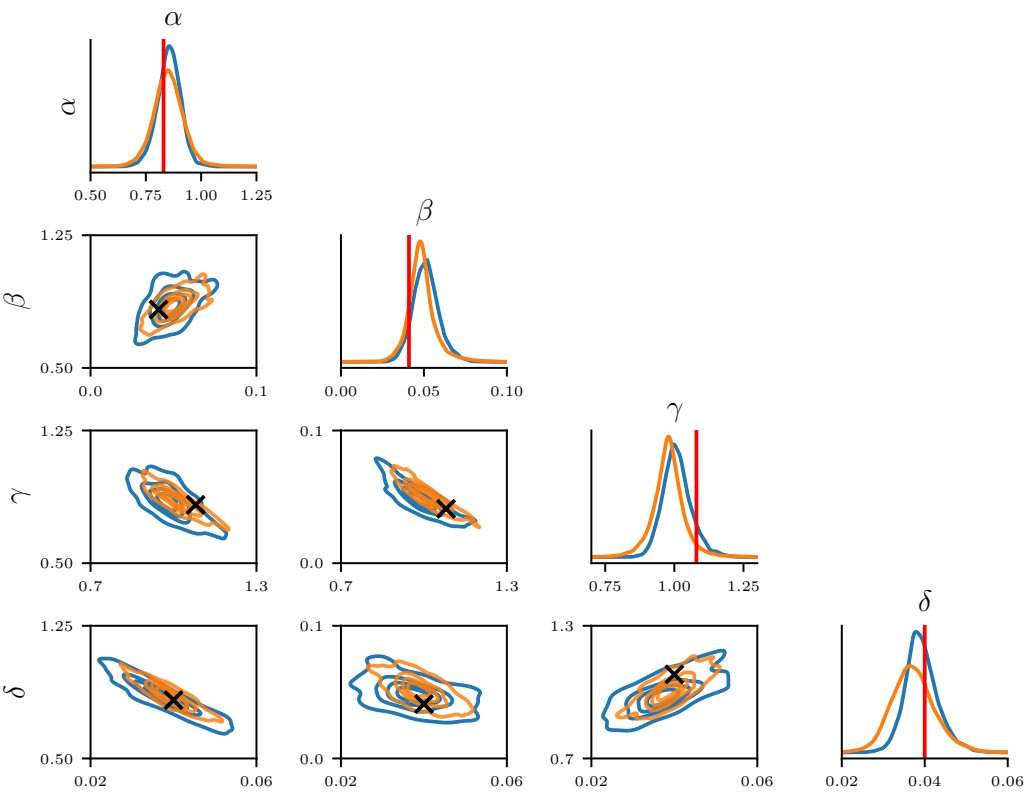

Figure 7: KDE plots of the samples on the Lotka-Volterra system, using the settings described in Section 7. Plots include one-dimensional KDEs on the diagonal, as well as all two-dimensional pairs. In all plots, samples from MCMC are drawn in orange, and samples from our method (COT-FM) are indicated in blue. The true unknown parameters are indicated by the red vertical line in the diagonal plots, or the black x in the off-diagonal plots.

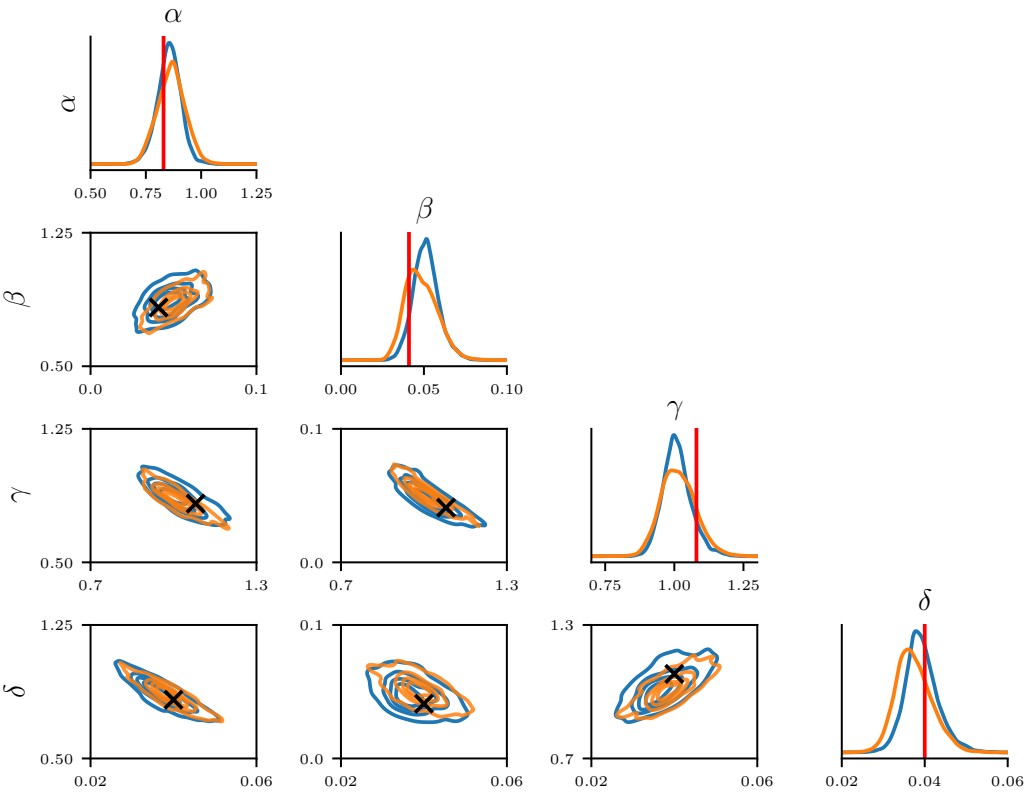

Figure 8: KDE plots of the samples on the Lotka-Volterra system, using the settings described in Section 7. Plots include one-dimensional KDEs on the diagonal, as well as all two-dimensional pairs. In all plots, samples from MCMC are drawn in orange, and samples from PCP-Map are indicated in blue. The true unknown parameters are indicated by the red vertical line in the diagonal plots, or the black x in the off-diagonal plots.

Lotka-Volterra Samples: COT-Flow

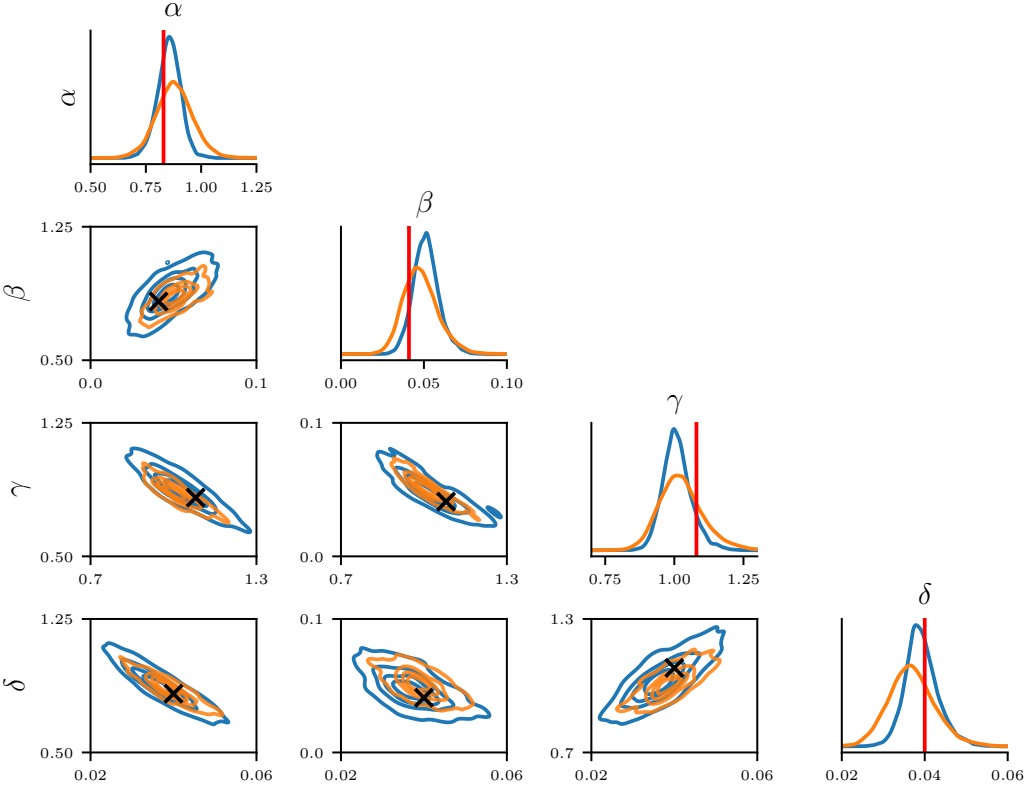

Figure 9: KDE plots of the samples on the Lotka-Volterra system, using the settings described in Section 7. Plots include one-dimensional KDEs on the diagonal, as well as all two-dimensional pairs. In all plots, samples from MCMC are drawn in orange, and samples from COT-Flow are indicated in blue. The true unknown parameters are indicated by the red vertical line in the diagonal plots, or the black x in the off-diagonal plots.

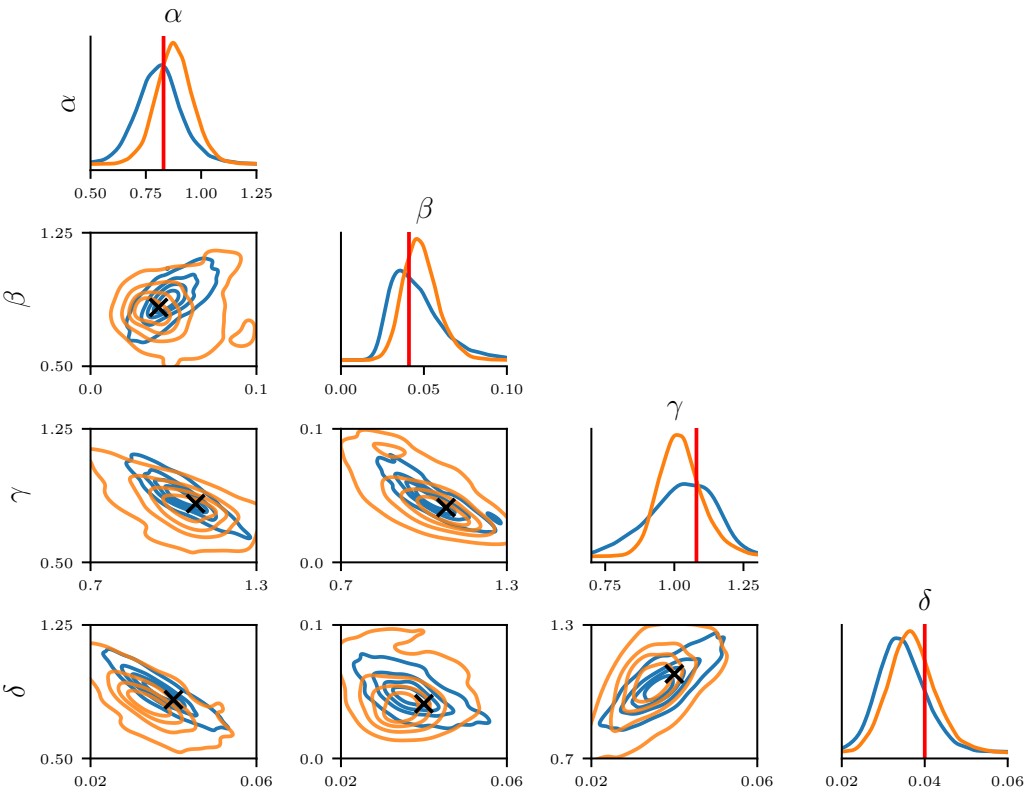

Figure 10: KDE plots of the samples on the Lotka-Volterra system, using the settings described in Section 7. Plots include one-dimensional KDEs on the diagonal, as well as all two-dimensional pairs. In all plots, samples from MCMC are drawn in orange, and samples from flow matching (FM) are indicated in blue. The true unknown parameters are indicated by the red vertical line in the diagonal plots, or the black x in the off-diagonal plots.

## F.3 Inverse Darcy Flow

**Dataset.** The training and test datasets are generated following the same procedure as Hosseini et al. [2023]: pressure fields $u$ are sampled from a Gaussian process with Matérn kernel having $\nu = 3/2$ and lengthscale $\ell = 1/2$, on a regular $40 \times 40$ grid. The parameters are then exponentiated and used to simulate the permeability fields $p$ from the forward model $\mathfrak{F}$ solving the Darcy flow PDE, using FEniCS [Alnæs et al., 2015]. Stochasticity arises from adding Gaussian noise to the permeability fields, obtaining $y = \mathfrak{F}(u) + \epsilon$, $\epsilon \sim \mathcal{N}(0, \sigma^2 I)$. For our experiments we observe $y$ on a $100 \times 100$ grid, and we use $\sigma = 2.5 \times 10^{-2}$. We note that this level of noise is quite considerable, as it accounts for roughly $60\%$ of the variability in the $y$. Figure 11 showcases a data point for reference. Our source and target training sets contain $1 \times 10^4$ samples each, and our test set comprises $5 \times 10^3$ samples. We remark that although $y$ and $u$ are observed on a grid their resolution does not need to be fixed, allowing for training at different resolutions.

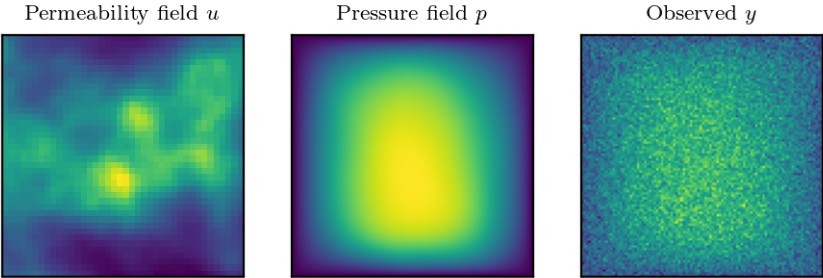

Figure 11: Example of one random data point from the Darcy flow dataset.

**Models.** In order to make learning feasible in infinite-dimensional Hilbert spaces, we adapt the architecture of a Fourier Neural Operator (FNO) [Li et al., 2020] from the `neuraloperator` package [Kovachki et al., 2021] to accommodate for conditioning information observed at an arbitrary resolution. We do so by introducing a projection layer mapping the conditioning information to match the hidden channels of the input lifting block, and a pooling operation to project to the input dimensions. The two are then concatenated and passed through an `FNOBlock` mapping from $(2 \times \texttt{hidden\_channels}) \times \texttt{input\_dim}$ to $\texttt{hidden\_channels} \times \texttt{input\_dim}$, before following the original architecture. For all of the models in consideration, we fix the architecture to be have $\texttt{hidden\_channels} = 64$, $\texttt{projection\_channels} = 256$, and 32 Fourier modes. We train each model for 1500 epochs, and hyperparameters for each architecture are selected as follows:

- WaMGAN [Hosseini et al., 2023]: using an adaptation to the FNO architecture of the original code[3], we perform a grid search as detailed in Table 5. We found the training procedure to be rather unstable, and for this reason we checkpoint the model every 100 epochs and report the results for the best performing model at its best checkpoint. We found this to be a model with learning rate $1 \times 10^{-4}$, 2 full critic iterations, and monotone penalty of $1 \times 10^{-3}$. The gradient penalty parameter did not seem to significantly affect performance on the test set, and was set to 5.

- FFM [Kerrigan et al., 2024]: the learning rate is fixed to $5 \times 10^{-4}$, and the covariance operator $C$ is set to match that of the prior, but rescaled by a factor of $\sigma = 1 \times 10^{-3}$. We use the code from the original repository[4].

- COT-FFM: we set $\epsilon = 1 \times 10^{-5}$ in the cost function used to build the COT plan. The learning rate and $C$ are chosen to be the same as FFM. In order to build COT couplings, we take the source measure to be the product measure $\pi_{\#}^Y \eta \times \mathcal{N}(0, C)$. Approximate couplings are obtained on minibatches of size 256.

It should be noted that in any scenario where the source and the target $U-$marginals are identical, using the OT coupling would yield the identity mapping as the optimal vector field minimizing (12). Hence, the OT-CFM model [Tong et al., 2023] is inapplicable here.

---

[3]https://github.com/TADSGroup/ConditionalOT2023
[4]https://github.com/GavinKerrigan/functional_flow_matching

Table 5: Hyperparameter search space for WaMGAN

| Parameter | Search Space |
|---|---|
| Learning rate | $\{1 \times 10^{-3}, 5 \times 10^{-4}, 1 \times 10^{-4}\}$ |
| Full critic iter. | $\{2, 5, 10\}$ |
| Monotone penalty | $\{1 \times 10^{-3}, 5 \times 10^{-2}, 1 \times 10^{-1}\}$ |
| Gradient penalty | $\{1, 5, 10\}$ |

**Sampling.** The resulting amortized sampler, denoted for simplification by the mapping $(y, u_0) \mapsto u_1 = \tilde{T}_U(y, u_0)$, will parameterize an approximate posterior measure. Notice that, in contrast to classical variational inference techniques, no distributional assumptions are made about the approximate posterior. In turn, integrals are obtained numerically by Monte Carlo sampling $K$ samples from the prior, resulting in the approximation

$$\nu^y(f) \approx \int f \, \mathrm{d}\delta_{\tilde{T}_U(y, u_0)} \, \mathrm{d}\mathcal{N}(0, C) \approx \frac{1}{K} \sum_{k=1}^{K} f(\tilde{T}_U(y_k, u_{0,k})), \quad \{u_{0,k}\}_{k=1}^{K} \stackrel{\text{i.i.d.}}{\sim} \mathcal{N}(0, C). \quad (105)$$

# G   Minibatch COT

In this section, we perform an additional experiment demonstrating that our COT-FM is able to obtain good approximations of the true COT map even when trained on relatively small minibatches.

We work on $Y = U = \mathbb{R}$ and use a standard Gaussian source and a Gaussian target distribution with covariance $\rho = 0.75$. These are chosen so that we may evaluate, in closed-form, the true conditional 2-Wasserstein distance, as detailed in Section C. We then train our COT-FM method using various batch sizes, and measure the resulting model's conditional 2-Wasserstein distance by sampling $10,000$ points from the source distribution, flowing each source point along the model's learned vector field, and computing the resulting squared distance to the corresponding terminal point.

In Figure 12, we plot the resulting deviation from the true value of $W_2^{\mu, 2}$ as a function of batch size. We see that even at a relatively small batch size of $16$, the resulting error in the (squared) distances is less than 10% of the true value ($\approx 0.678$). While this experiment is only feasible on synthetic data (as we must compute the true distance), it nonetheless demonstrates that even with small batches we may recover the true distance.

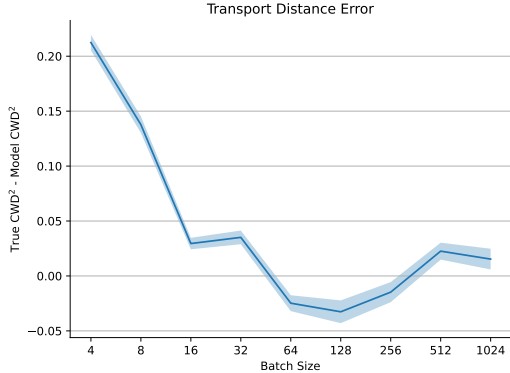

Figure 12: Error in the squared transportation distance as a function of batch size. Values closer to zero indicate a better approximation of the COT cost. For fairly small batch sizes ($> 16$) the magnitude of the error is stable and relatively small. Shading indicates one standard deviation computed over ten random evaluation sets.

