# OpenReview forum: "Dynamic Conditional Optimal Transport through Simulation-Free Flows"
_NeurIPS.cc/2024/Conference — NeurIPS 2024 poster_

### Official Review · Reviewer_i1c7 · 2024-07-07

**Soundness:** 2
**Presentation:** 3
**Contribution:** 2
**Rating:** 4
**Confidence:** 3

**Summary:**

This paper introduces COT-FM, a generalization of the Flow Matching model for conditional generation. Specifically, this paper investigates the Conditional Wasserstein Space, a space of joint probability measures on $Y \times U$ with fixed $Y$-mariginals $\mu$. This paper proves that an absolutely continuous path in the Conditional Wasserstein Space can be generated by a triangular vector field. Based on this characterization, the COT-FM is proposed as a Flow Matching model that employs a triangular vector field on the $Y \times U$.

**Strengths:**

- This paper presents theoretical analysis of the Conditional Wasserstein Space, such as the characterization of the absolutely continuous path and the conditional generalization of the Benamou-Brenier Theorem.
- This paper proposes a Flow Matching model for conditional generation.
- This paper is easy to follow.

**Weaknesses:**

- Whether the COT-FM can recover the dynamic optimal transport requires further clarification.
- Please see the Questions Section below.

**Questions:**

- I would like to clarify the connection between Section (4, 5) and Section 6. It appears that Section (4,5) establish the existence of an absolutely continuous path between two arbitrary measures in the Conditional Wasserstein Space, which can be generated by a triangular vector field. In this context, Section 6 introduces a triangular vector field parametrization to the Flow Matching model. Hence, Section (4,5) justify the triangular parametrization of the standard Flow Matching model within the Conditional Wasserstein Space. Is this correct?
- I am curious whether COT-FM can recover the dynamic optimal transport within the Conditional Wasserstein Space, as mentioned in Line 256. Proposition 3.4 in [Tong et al., 2023] addresses the standard Wasserstein Space case. Could you provide clarification on how this applies to the Conditional Wasserstein Space?
- Table 1 presents the W2 and MMD distances between the joint distributions. Could you also provide the W2 and MMD results between the conditional distributions?
- I am curious about the significance of minibatch optimal coupling for COT-FM, in Lines 257-265. Could you provide the COT-FM results using independent coupling?

**Limitations:**

-	The authors addressed the potential negative societal impact of their work.

---

> ### Author Rebuttal · Authors · 2024-08-05
>
> We are grateful to the reviewer for their constructive feedback and suggestions.
>
> > I would like to clarify the connection between Section (4, 5) and Section 6 [...] Section (4,5) justify the triangular parametrization of the standard Flow Matching model within the Conditional Wasserstein Space. Is this correct?
>
> Yes, your understanding is correct. We agree that the relationship between our theoretical and methodological contributions could be better clarified, and we will update our paper to make this more clear.
>
> Here, we provide some additional context. Inspired by the empirical success of OT couplings in unconditional generative modeling [Tong 23, Pooladian 23], we sought to employ similar techniques for conditional generative modeling. However, prior to our work, the theory of conditional optimal transport was not sufficiently developed to justify this approach. Thus, a major contribution of our work (Sections 4, 5) is the development of such a theory which provides a principled foundation for COT-FM (Section 6). However, we believe that our theoretical contributions provide a foundation for COT in general, beyond only our proposed COT-FM method.
>
> We additionally would like to clarify that COT-FM requires more than simply using triangular vector fields in the standard flow matching model. In particular, COT-FM requires one to additionally compute the static COT couplings, as discussed in lines 267-265.
>
>
> > I am curious whether COT-FM can recover the dynamic optimal transport within the Conditional Wasserstein Space, as mentioned in Line 256. Proposition 3.4 in [Tong et al., 2023] addresses the standard Wasserstein Space case. Could you provide clarification on how this applies to the Conditional Wasserstein Space?
>
> Our COT-FM indeed recovers the dynamic COT paths in the limit of zero smoothing. We write in Line 256 that this follows from a “pointwise application of [Tong et al., 2023, Prop. 3.4]”. Here, by pointwise, we mean that we may apply the result of [Tong 23] for a single, fixed $y$ to recover the optimal path conditioned on $y$ -- and thus, over all possible values of $y$.
>
> Somewhat more precisely, Prop 3.4 of [Tong 23] shows in the unconditional setting that $v_t^\sigma(u) \to v_t(u)$ as $\sigma \to 0$, where $v_t^\sigma(u)$ is the smoothed flow matching vector field, and $v_t(u)$ is the dynamic OT vector field. In the *conditional* setting, our vector fields are of the form $v_t^\sigma(y, u)$. However, if we fix $y$, this vector field can be viewed as a vector field purely over the $U$ component, thanks to the triangularity assumption. This allows us to directly apply Prop 3.4 of [Tong 23] -- in particular, because the COT problem is essentially a collection of unconditional OT problems, one for each fixed $y$. See e.g. [Hosseini 23].
>
> We agree that this point could have been explained more clearly, and we will update our paper with a complete, formal proof of this result.
>
> In our global response, we additionally conduct an experiment to measure the degree to which the COT paths are optimal. Overall, we find that the paths learned by COT-FM are close to optimal, even when using the minibatch approximation.
>
> > Table 1 presents the W2 and MMD distances between the joint distributions. Could you also provide the W2 and MMD results between the conditional distributions?
>
> We would like to clarify that we measure the joint metrics for practical reasons. Namely, to measure the conditional MMD or Wasserstein distances, we would require the ability to sample many values of $u \sim \nu (u \mid y)$ from the data distribution, conditioned on a given $y$. However, in many practical instances (including the Lotka-Volterra, Darcy Flow, and many of the 2D datasets), this is not possible, as the data generation mechanism either directly produces sample $(y, u) \sim \nu(y, u)$ from the joint distribution (as in the 2D data) or produces samples $(y, u) \sim \nu(u) \nu(y \mid u)$ from a prior over $u$ and a forward model $\nu(y \mid u)$ (as in the Lotka-Volterra and Darcy Flow problems).
>
> Moreover, even in cases where we can indeed sample $u \sim \nu(u \mid y)$, we found that estimators of the conditional distances have prohibitively high variance, as it is in essence a nested sampling problem.
>
> Thus, some approximate measure of model fit is necessary. We believe the joint metrics are a reasonable proxy since, by Proposition 1, any triangular mapping which couples the joint distributions necessarily couples the conditional distributions.
>
> We thank the reviewer for pointing out this difference, and we will use the additional space afforded by a camera-ready version of our paper to discuss these challenges and justifications.
>
>
> > I am curious about the significance of minibatch optimal coupling for COT-FM, in Lines 257-265. Could you provide the COT-FM results using independent coupling?
>
> Please see our global response for a discussion regarding the role of minibatches in our method. We would additionally like to point out that we do indeed compare our method against flow matching with an independent coupling. This is “FM” in Table 1 and Table 2, and “FFM” in Table 3. Overall, there is a clear gap between COT-FM and FM with independent couplings across all experiments considered in our work.
>
> We agree, though, that this could have been better explained in our submission. We will update our paper with a more clear explanation of the baseline flow matching model.
>
> 1. Improving and Generalizing Flow-Based Generative Models with Minibatch Optimal Transport. Tong et al., 2023.
> 2. Multisample Flow Matching: Straightening Flows with Minibatch Couplings. Pooladian et al., 2023.
> 3. Conditional Optimal Transport on Function Spaces. Hosseini et al., 2023.

---

> > ### Comment · Reviewer_i1c7 · 2024-08-11
> >
> > I appreciate the author for their clarifications. However, I still believe that additional quantitative evaluation as a conditional generative model would provide more solid support for this work. Hence, I will maintain my current score.

---

### Official Review · Reviewer_yVF4 · 2024-07-10

**Soundness:** 4
**Presentation:** 4
**Contribution:** 3
**Rating:** 8
**Confidence:** 5

**Summary:**

This paper provides a theory for conditional optimal transport (as defined by the authors), followed by numerical simulations. Among their contributions, the authors put forth theory for the geometry of the conditional Wasserstein space (where analogous quantities of e.g., the McCann interpolation, hold). This is due to the geometry given by triangular vector fields that are studied. Their proposed algorithm is based off Flow Matching [Lipman et al. 2023], and achieves strong numerical performance against the other baseline algorithms.

**Strengths:**

This paper has many strengths! It is well-written, fits nicely in the conference format without omitting many details, and has appropriate experiments. The proposed methodology is also quite elegant, and circumvents many issues other methods face.

**Weaknesses:**

N/A :)

**Questions:**

I am using this space for comments and suggestions as well as questions.

- Is there a clear way to choose the $\epsilon$ parameter for the COT Flow Matching? Any heuristics whatsoever? This appears to be a bottleneck to making this methodology fully practical
- Convergence of the $\epsilon\to 0$ limit of the proposed OT map for the twisted cost is originally due to Carlier et al (2010) --- I would argue that the recent results by Hosseini et al. (2023) are extensions of this older result.
- When citing flow matching throughout the draft, it would be equitable to also cite Liu et al. (2023) alongside Lipman et al. 2023 and Albergo et al. (2023). Same goes for Pooladian et al. (2023) --- should be cited alongside Tong et al. (2023) (in e.g., Section 6)
- Is equation (9) not due to the original flow matching papers?
- For equation (6): I have never heard anyone say the equation should be "understood distributionally". Maybe consider "in the sense of distributions"
- Stylistic comment: Maybe omit "unconditional" from the title of Appendix A? This is not really used

@article{carlier2010knothe,
  title={From Knothe's transport to Brenier's map and a continuation method for optimal transport},
  author={Carlier, Guillaume and Galichon, Alfred and Santambrogio, Filippo},
  journal={SIAM Journal on Mathematical Analysis},
  volume={41},
  number={6},
  pages={2554--2576},
  year={2010},
  publisher={SIAM}
}

@article{liu2022flow,
  title={Flow straight and fast: Learning to generate and transfer data with rectified flow},
  author={Liu, Xingchao and Gong, Chengyue and Liu, Qiang},
  journal={arXiv preprint arXiv:2209.03003},
  year={2022}
}

---

> ### Author Rebuttal · Authors · 2024-08-05
>
> We thank the reviewer for their detailed feedback and positive review! We have updated our paper to incorporate the various suggested references and stylistic edits.
>
> > Is there a clear way to choose the $\epsilon$ parameter for the COT Flow Matching? Any heuristics whatsoever?
>
> This is an interesting question -- while we do not have any rigorous results in this direction, we can perhaps give some intuition. Ideally, one would choose a small value of $\epsilon$ -- as suggested by the results of Carlier [2010], with unlimited data this would indeed recover a triangular map. However, we expect there to be dependence between $\epsilon$ and the size of the dataset (or batch size, if using minibatch couplings). Loosely speaking, $\epsilon$ controls the tradeoff between triangularity and coupling nearby points; small values of $\epsilon$ encourage the maps to be triangular at the cost of a potentially large distance in the $U$ component, whereas large values of $\epsilon$ allow greater violations of triangularity while coupling points that are nearby in $U$ distance.
>
> Moreover, we would expect this $U$ distance to be larger for small batches of data, since there may not be a source point with the same (or nearby) $Y$ values as a target point. Thus we would expect that $\epsilon$ and the batch size should be negatively correlated.
>
> However, in practice, $\epsilon$ is a nuisance parameter which can be tuned just as any other parameter in the learning process. In practice, we tuned $\epsilon$ through a grid search, measuring the loss on held-out validation data.
>
> We will include a discussion of these notions in an updated camera-ready version of our paper.
>
> > Is equation (9) not due to the original flow matching papers?
>
> Yes, essentially -- a special case of Equation (9) appears in the early work of [Theorem 2, Lipman 23], where the coupling depends only on the target variable $x_1$. An equivalent loss appears in Proposition 1 of [Albergo 23A], but still using an independent coupling. Theorem 3.2 of [Tong et al., 2023] extends this loss to general couplings, and [Albergo 23B, Pooladian 23] propose a similar use of general couplings. We will update the citation in our paper to better reflect this.
>
> 1. Flow Matching for Generative Modeling. Lipman et al., 2023.
> 2. Building Normalizing Flows with Stochastic Interpolants. Albergo et al., 2023(A).
> 3. Stochastic Interpolants with Data-Dependent Couplings. Albergo et al., 2023(B).
> 4. Improving and Generalizing Flow-Based Generative Models with Minibatch Optimal Transport. Tong et al., 2023.
> 5. Multisample Flow Matching: Straightening Flows with Minibatch Couplings. Pooladian et al., 2023.

---

### Official Review · Reviewer_hqSP · 2024-07-12

**Soundness:** 3
**Presentation:** 2
**Contribution:** 2
**Rating:** 5
**Confidence:** 3

**Summary:**

This paper characterizes dynamical conditional optimal transport (COT). It generalizes the Benamou-Brenier theorem to dynamical COT. The authors then propose conditional flow matching and apply it to synthetic data.

**Strengths:**

- The paper successfully extends the Benamou-Brenier theorem to the context of dynamical conditional optimal transport.
- The paper is easy to follow.

**Weaknesses:**

- The paper appears to be extremely similar to [1] both in theoretically and empirically. Particularly, Theorem 18, which discusses a Benamou-Brenier-like formula and it applies COT with flow matching as like this work. I would like to ask authors to discuss the difference with result in [1].

- The paper discusses dynamical COT in Sections 4-5. However, in Section 6, the authors propose the Conditional OT Flow Matching (COT-FM) method. This method solves dynamic COT only when the given joint coupling is the solution of COT. In other words, optimal coupling should be given for COT-FM algorithm to solve dynamic COT problem. In OT literature, the most of the application aims to find optimal coupling (rather than given), hence, this algorithm can be applied only in very restricted situation. Thus, the application as an OT method is extremely limited.

- Moreover, in the conducted experiments, the given pairs are not the solutions to COT (only mini-batch sense). Therefore, the experiments seem to address conditional generation rather than conditional optimal transport. It is unclear if the experimental settings are appropriate for the subject of the paper. Moreover, the experiments were conducted on very small datasets.

[1] Conditional Wasserstein Distances with Applications in Bayesian OT Flow Matching (arxiv, v1 released in March, 2024)

**Questions:**

- In line 51, it is discussed that "COT-FM ... interpolates between an arbitrary source and target distribution via a geodesic in the conditional Wasserstein space". Does it mean that FM model learn geodesics?

**Limitations:**

The limitation is discussed in Weakness section.

---

> ### Author Rebuttal · Authors · 2024-08-05
>
> We thank the reviewer for their feedback!
>
> > The paper appears to be extremely similar to [1] [...] I would like to ask authors to discuss the difference with result in [1].
>
> We first would like to remind the reviewer of the [NeurIPS concurrent work policy](https://neurips.cc/Conferences/2024/PaperInformation/NeurIPS-FAQ). Given that [Chemseddine 24] first appeared on 27 March 2024 and the NeurIPS submission deadline was 22 May 2024, this work counts as concurrent. This work was developed independently and concurrently with our submission, which we briefly mention in lines 78-80. However, there are several differences. An updated version of our paper will elaborate on these.
>
> 1. [Chemseddine 24] work in finite-dimensional Euclidean spaces. Our theory is applicable in more general spaces -- including many infinite-dimensional function spaces. This necessitates additional techniques in our proofs (particularly Lemma 1 Theorem 4). Theorem 18 of [Chemseddine 24] can be viewed as a special case of our Theorem 5 under a finite-dimensional assumption.
>
> 2. Prop 6 [Chemseddine 24] shows existence of vector fields generating a path of measures when the path is induced by an optimal plan. We show a stronger result in Theorem 3, where we characterize **all** absolutely continuous curves, rather than only those induced by an optimal plan. We also prove a converse of this statement in Theorem 4 which does not appear in [Chemseddine 2024].
>
> > The paper discusses dynamical COT in Sections 4-5. [...] the experiments seem to address conditional generation rather than conditional optimal transport.
>
> As discussed in our abstract and introduction, we are indeed primarily interested in applications in conditional generative modeling. Inspired by the success of OT couplings in unconditional generative modeling [Tong 23, Pooladian 23], we sought to employ similar techniques. However, prior to our work, the theory of conditional optimal transport was not sufficiently developed to justify this approach. Thus, a major contribution of our work is the development of such a theory. We believe that our theoretical contributions provide a foundation for COT in general, beyond only our proposed COT-FM method.
>
> We agree with the reviewer that this point could have been made more clear, and we will update our paper to discuss the relationship between Sections 4-5 and Section 6.
>
> > This method solves dynamic COT only when the given joint coupling is the solution of COT. [...] the application as an OT method is extremely limited.
>
> Your understanding is correct -- however, we respectfully disagree that our method has limited applicability.
>
> First, approximating a static COT coupling is relatively straightforward [Carlier 10]. We discuss this in lines 257-265. However, **such a coupling is necessarily empirical** -- that is, the coupling is only between the observed data, and it is not straightforward to generalize to unseen data. Although our method requires these empirical static couplings at training time, we may apply our method to simulate the transport for unseen data. This is particularly useful when one is interested in modeling the transport for large datasets, for which the standard approaches to COT would be prohibitively expensive in terms of memory.
>
> Second, as mentioned previously, we are primarily interested in applications in generative modeling. As demonstrated by our experiments, computing the static COT couplings is not a prohibitive step in solving conditional generation tasks. This is further emphasized by the success of OT based flow-matching algorithms for unconditional generation [Tong 23, Pooladian 23].
>
> > It is unclear if the experimental settings are appropriate
>
> As discussed above, our methodological contributions are focused on conditional generative modeling. Our experiments are aligned with this aim -- in Section 7, we demonstrate that our proposed method, COT-FM, obtains strong empirical performance across a range of conditional generation tasks. We would be happy to hear suggestions for additional experiments that would further improve the paper.
>
> > the experiments were conducted on very small datasets.
>
> In Appendix F, we discuss the size of our datasets -- namely, we train on 20,000 datapoints for our 2D experiments and 10,000 datapoints for our Lotka-Volterra and Darcy flow experiments. We respectfully disagree that these datasets are too small, as many real-world applications of inverse problems have datasets of roughly this magnitude. Moreover, we believe our method will scale to larger datasets, as the additional overhead as compared to e.g. standard flow matching is not prohibitively expensive.
>
> > Does it mean that FM model learn geodesics?
>
> Yes -- the path of distributions modeled by COT-FM is a geodesic in the conditional Wasserstein space, in the sense of Theorem 1.
>
> Informally, Theorem 5 shows these geodesics are induced by triangular vector fields. Moreover, we show in Theorem 2(a) an optimal COT mapping induces a geodesic in this space, where the vector field producing this geodesic is given in Theorem 2(b). The vector fields in Theorem 2(b) are precisely those we use to learn COT-FM in Equation (7). While there is some necessary additional smoothing, in the limit of zero covariance we recover the true geodesics in theory.
>
> We briefly discuss this in Lines 254-256 of our submission, but we agree that this could have been explained more clearly. If accepted, the camera-ready version of our paper will include a more extensive discussion.
>
>
> 1. Conditional Wasserstein Distances with Applications in Bayesian OT Flow Matching. Chemseddine et al., 2024
> 2. Improving and generalizing flow-based generative models with minibatch optimal transport. Tong et al., 2023.
> 3. Multisample Flow Matching: Straightening Flows with Minibatch Couplings. Pooladian et al., 2023.
> 4. From Knothe's transport to Brenier's map and a continuation method for optimal transport. Carlier et al., 2010

---

> > ### Comment · Reviewer_hqSP · 2024-08-11
> >
> > I appreciate the author for the clarification. I agree that the main contribution of this work is the development of the dynamical conditional optimal transport theory, and it is quite novel. I also agree that the COT map can be obtained as the mini-batch size approaches infinity (as discussed in [Tong, 2023]). The additional experiments, which demonstrate that it is possible to approximate the actual COT with large batch sizes, enhance the soundness of the approach. Although there are some aspects of the methodology that might be open to discussion, considering the theoretical contributions and the additional experiments, I would like to raise my score to 5.

---

> ### Comment · Area_Chair_d3C8 · 2024-08-06
> **Comparison to previous work**
>
> Dear reviewer hqSP,
>
> Thank you for your work evaluating this submission.
>
> The internal policy ([link](https://neurips.cc/Conferences/2024/PaperInformation/NeurIPS-FAQ)) states that work appearing two months or less before the deadline - which is the case of the work you refer to - is considered concurrent, and that authors should not be expected to compare to such work.
>
> If part of your review was based on this, it will be possible to update it during the discussion phase.

---

### Official Review · Reviewer_QX6V · 2024-07-13

**Soundness:** 4
**Presentation:** 3
**Contribution:** 3
**Rating:** 6
**Confidence:** 2

**Summary:**

This work first extends conditional optimal transport theory to the dynamical setting. Then a flow-matching model is proposed to approximate these flows with a simulation free training objective. This is then applied in several conditional generation tasks including two Bayesian inverse problems. Triangular optimal transport maps are used in combination with the dynamic Brenier-Benamou formulation of standard optimal transport.

**Strengths:**

- This work combines ideas from conditional optimal transport and optimal transport flow matching to create a new approach for Bayesian inverse problems and likelihood-free inference.
- The results effectively demonstrate how this approach can be used in a variety of settings.
- The work is well written and fairly clear

**Weaknesses:**

- Only applied to relatively low dimensional problems.
- It might be useful to clearly highlight the power and generalization of this work over a “simple” conditional optimal transport formulation which simply conditions on a single-class variable. e.g. https://github.com/atong01/conditional-flow-matching/blob/main/examples/images/conditional_mnist.ipynb

    While it is clear from a deep enough reading I suggest the authors might want to highlight these differences for potentially broader appeal. This might be done through an algorithm box or other presentation.

**Questions:**

It would be good to know empirically how far from optimal the learned transport maps are, as this is a known limitation of miqibatch-based approaches.

Comment: I would suggest citing work on rectified flows as a concurrent invention of flow matching.
https://arxiv.org/abs/2209.03003

**Limitations:**

Adequately discussed.

---

> ### Author Rebuttal · Authors · 2024-08-05
>
> We thank the reviewer for their valuable feedback!
>
> > It would be good to know empirically how far from optimal the learned transport maps are, as this is a known limitation of mi[n]ibatch-based approaches.
>
> We thank the reviewer for pointing this out. Please see our global response for a discussion regarding minibatches and COT. Overall, we find that our method COT-FM can recover nearly optimal transport distances even when trained at small batch sizes.
>
> > Only applied to relatively low dimensional problems.
>
> This is a fair point. However, we would like to emphasize that our most significant contributions are to the theory of conditional optimal transport, laying the foundations for future applications.
>
> Moreover, many of the applications we have in mind are inverse problems, which are often fairly low-dimensional, e.g. recovering the parameters of an ODE as in our Lotka-Volterra experiment in Section 7. We also demonstrate our method on image data for the Darcy Flow inverse problem. While this is still relatively low-dimensional (at a grid size of 40x40), this was largely due to the computational cost of solving PDEs at high resolutions, which is necessary for both producing the training data as well as running MCMC as a point of comparison.
>
> These results suggest that the method should scale to higher dimensional problems, which is an interesting avenue for future work.
>
> > It might be useful to clearly highlight the power and generalization of this work over a “simple” conditional optimal transport formulation which simply conditions on a single-class variable.
>
> Thank you for this suggestion -- we agree that this could be more clearly explained. We elaborate here on the key differences.
>
> When the conditioning variable takes values in a finite set, like a class label, the conditional optimal transport problem becomes much simpler. This is because, in this scenario, we typically have many observations (in the $U$ space) for any single, given conditioning variable $y$ -- e.g., many images all coming from the same class. In this case, one can simply solve the usual optimal transport problem for each class independently.
>
> However, we are largely interested in problems where we only have a single observation $u$ for each given $y$ -- this is the case, for instance, in inverse problems. Here, it is impossible to solve the optimal transport problem directly for each $y$ independently as there is simply not enough data.
>
> We describe this briefly in lines 98-106, but we will make this point more clear. We will include pseudocode in our camera-ready submission to highlight the differences between our method and existing work.
>
>
> > I would suggest citing work on rectified flows as a concurrent invention of flow matching.
>
> We thank the reviewer for the reference -- we are aware of rectified flows, and we have updated our paper with a short discussion of this work and how it relates to our submission and flow matching more broadly.

---

> > ### Comment · Reviewer_QX6V · 2024-08-11
> >
> > I thank the authors for their clarifications and additional experiments. I still believe this makes a potentially useful theoretical contribution but with limited empirical evaluation and therefore maintain my score.

---

### Author Rebuttal · Authors · 2024-08-05

# Summary
We would like to thank the reviewers for their detailed and valuable feedback. We are encouraged to hear that the reviewers found our theoretical contributions to be a strength (QX6V hqSP, i1c7, yVF4) and that the submission is well-written (QX6v, hqSP, yVF4, i1c7). We are also glad the reviewers found our proposed method “quite elegant” (yVF4), with experimental results that clearly demonstrate the strengths of the approach (QX6v, yVF4).

# Minibatch COT
Several reviewers had questions regarding the role of minibatches in our setup. While the use of minibatches is needed to scale the method to large datasets, we believe that this limitation is not prohibitive in practical adaptations of these methods. Our experiments (Tables 1, 2, 3) demonstrate that even with a minibatch approximation, COT-FM outperforms standard flow matching (without the use of COT).

The precise relationship between minibatch OT and the standard OT problem is an area of active research. However, there have been some promising results, at least in the unconditional setting. For instance, [Bernton 19] show that minimizers of the batched Wasserstein distance converge to minimizers of unbatched Wasserstein distance (as the batch size grows), and [Sommerfeld 19] establish non-asymptotic bounds between the Wasserstein distance and its minibtach approximation. Studying these relationships for the conditional OT problem is an open, challenging problem and an exciting direction for future work.

To further investigate the role of minibatches, we conducted an additional experiment. In this experiment, we use a standard Gaussian as our source distribution, and a Gaussian with covariance $\rho = 0.75$ as the target. We choose these distributions as the COT distance is available in closed form, as we prove in Section 4 / Appendix C. We then train our proposed COT-FM model on a range of different batch sizes (leaving all other hyperparameters fixed except for the learning rate). We then measure the degree to which the transport learned by COT-FM is optimal. Figures for our preliminary results are contained in the attached response.

In Figure 1, we plot the difference between the true value of the squared COT distance $W_2^{\mu, 2}$, and the squared transport cost resulting from the trained COT-FM model. This learned cost is estimated by sampling 10,000 points from the source distribution, flowing each point along the model’s learned vector field, and computing the Euclidean distance between the initial and terminal location of each point. We see that even for relatively small batch sizes, the transport costs from our learned COT-FM model closely match the ground-truth cost.

In Figure 2, we additionally measure the degree to which the learned dynamic COT paths are optimal. To do so, we measure the difference between the path energy of the model $\int_0^1 \int |v_t^\theta| d p_t d t$ and the known ground-truth distance $W_2^{\mu, 2}$. By Theorem 5 of our work, these two quantities should be equal if the learned dynamic paths are optimal. We find that the deviation decreases for larger batch sizes, but that the magnitude of the deviation is fairly small relative to the distance.

Overall, these two results indicate that COT-FM is able to obtain good approximations of both the dynamic and static COT solution, even when using minibatches. These results are in agreement with similar results in the unconditional case [Tong 23, Fatras 21]. We will conduct additional experiments with the datasets considered in the main submission to assess the degree of optimality in an updated version of our paper.

1. On parameter estimation with the Wasserstein distance. Bernton et al., 2019.
2. Improving Mini-batch Optimal Transport via Partial Transportation. Nguyen et al., 2021.
3. Optimal Transport: Fast Probabilistic Approximation with Exact Solvers. Sommerfeld et al., 2019.
4. Unbalanced minibatch Optimal Transport; applications to Domain Adaptation. Fatras et al., 2021.
5. Improving and generalizing flow-based generative models with minibatch optimal transport. Tong et al., 2023.

---

### Decision · Program_Chairs · 2024-09-25

**Decision:**

Accept (poster)

**Comment:**

The reviewers were quite split on this submission. Taking into account my own evaluation, as well as the expertise of the reviewer, I recommend to accept this paper.